# OFF-POLICY MULTI-STEP Q-LEARNING

## ABSTRACT

In the past few years, off-policy reinforcement learning methods have shown promising results in their application for robot control. Deep Q-learning, however, still suffers from poor data-efficiency which is limiting with regard to real-world applications. We follow the idea of multi-step TD-learning to enhance data-efficiency while remaining off-policy by proposing two novel Temporal-Difference formulations: (1) Truncated Q-functions which represent the return for the first $n$ steps of a target-policy rollout w.r.t. the full action-value and (2) Shifted Q-functions, acting as the farsighted return after this truncated rollout. We prove that the combination of these short- and long-term predictions is a representation of the full return, leading to the Composite Q-learning algorithm. We show the efficacy of Composite Q-learning in the tabular case and compare our approach in the function-approximation setting with TD3, Model-based Value Expansion and TD3($\Delta$), which we introduce as an off-policy variant of TD($\Delta$). We show on three simulated robot tasks that Composite TD3 outperforms TD3 as well as state-of-the-art off-policy multi-step approaches in terms of data-efficiency.

## 1 INTRODUCTION

In recent years, Q-learning (Watkins and Dayan, 1992) has achieved major successes in a broad range of areas by employing deep neural networks (Mnih et al., 2015; Silver et al., 2018; Lillicrap et al., 2016), including environments of higher complexity (Riedmiller et al., 2018) and even in first real world applications (Haarnoja et al., 2019). Due to its off-policy update, Q-learning can leverage transitions collected by any policy which makes it more data-efficient compared to on-policy methods. Deep Q-learning, however, still has a very high demand for data samples which is limiting with regard to robot applications.

One reason for the low data-efficiency is the long temporal horizon the reward signal has to propagate through. Data-efficiency of *on-policy* Temporal-Difference methods can be enhanced by the use of $n$-step returns, where a Monte Carlo rollout of length $n$ is combined with a bootstrap of the value function. To employ $n$-step returns in an *off-policy* setting, subtrajectories of the exploratory policy have to be stored. These stored multi-step returns, however, will differ from the true value of the target-policy. In order to benefit from $n$-step data, the replay buffer has to be restricted in size or $n$ has to be set to a small value to keep the samples close to the target-policy (Barth-Maron et al., 2018; Hessel et al., 2018). To avoid these problems, a dynamics model can be used for imaginary rollouts, the so-called *Model-based Value Expansion* (MVE) (Feinberg et al., 2018). Alternatively, the full return can be composed of value functions with increasing discount, an approach called TD($\Delta$) (Romoff et al., 2019). In this work, we define a model-free Temporal-Difference formulation which follows the idea of multi-step learning while remaining off-policy.

Our contributions are threefold. First, we introduce the *Composite Q-learning* algorithm. For its formulation, we define *Truncated Q-functions*, representing the return for the first $n$ steps of a target-policy rollout w.r.t. the full action-value. In addition, we introduce *Shifted Q-functions* which represent the farsighted return after this truncated rollout. Both are then combined in a mutual recursive definition of the Q-function for the final algorithm. Second, we evaluate MVE within TD3, leading to *MVE-TD3*. And third, we introduce *TD3($\Delta$)*, an extension of TD($\Delta$) to deep Q-learning. We discuss related work in Section 2, describe the theoretical background in Section 3 and define Composite Q-learning, MVE-TD3 and TD3($\Delta$) in Section 4. By breaking down the long-term return into a composition of several short-term predictions, our method increases data-efficiency which

we show in the tabular case and for three simulated robot tasks in Section 5. We then conclude in Section 6.

## 2 RELATED WORK

In order to correct for the deviation from the current target-policy, several methods suggest the use of Importance Sampling (Precup et al., 2000; 2001; Munos et al., 2016). Approaches based on Importance Sampling, however, can come with a vast increase in variance or can be of high cost and are not easily applicable to deterministic policies.

One way to remain off-policy in multi-step Q-learning is to get the Monte Carlo rollout on the basis of the current target-policy applied to a learned dynamics model (Feinberg et al., 2018; Buckman et al., 2018). Due to accumulating errors of single-step models, this can lead to severe stability issues in the Q-update, which is also the conclusion of Feinberg et al. (2018). The authors suggest to average all intermediate $i$-step returns to smoothen out the model error, the so-called TD-$k$ trick. In contrast to our work, Feinberg et al. assume to have access to the true reward function. In order to balance the length of the model-assisted rollout, Buckman et al. (2018) couple it with an uncertainty estimate from value function and model ensembles. We alleviate the problem of accumulating error by estimating a multi-step dynamics model *implicitly* via consecutive bootstrapping.

Most related to our approach is TD($\Delta$) (Romoff et al., 2019). Romoff et al. formalize a Bellman-operator over the differences between value functions of increasing discount values. Their approach is on-policy and can therefore benefit directly from $n$-step returns. We extend TD($\Delta$) to the off-policy case below. Q-learning can also be extended to a SARSA-like tree-backup called $Q(\sigma)$ (Asis et al., 2018; Hernandez-Garcia and Sutton, 2018). However, it is an open question how to adjust this idea to continuous action-spaces and deterministic policies. In the *Hybrid Reward Architecture* (HRA), van Seijen et al. (2017) suggest a decomposition of the reward and the estimation of value functions for each part of this decomposition which are then combined as an approximation of the full return. HRA addresses the problem of complex rewards and is thus complementary to our work focusing on long time scales. Concurrently to our work, Asis et al. (2019) introduced *Fixed Horizon TD-learning*, formalizing action-value functions for different horizons in time. In contrast to our work, however, Asis et al. define a consecutive bootstrapping formulation w.r.t. the target policies of the different truncated horizons and not the full return. A first general description of using intermediate predictions in Temporal Differences to restrict predictions to a fixed temporal horizon has been introduced in (Sutton, 1988).

## 3 BACKGROUND

We consider tasks modelled as Markov decision processes (MDP), where an agent executes action $a_t \in \mathcal{A}$ in some state $s_t \in \mathcal{S}$ following its stochastic policy $\pi$. According to the dynamics model $\mathcal{M}$ of the environment, the agent transitions into some state $s_{t+1} \in \mathcal{S}$ and receives scalar reward $r_t$. The agent aims at maximizing the expected long-term return:

$$\mathcal{R}^\pi(s_t) = \mathbf{E}_{a_{j \geq t} \sim \pi, s_{j > t} \sim \mathcal{M}} \left[ \sum_{j=t}^{T-1} \gamma^{j-t} r_j \,\middle|\, s_t \right], \tag{1}$$

where $T$ is the (possibly infinite) temporal horizon of the MDP and $\gamma \in [0, 1]$ the discount factor. It therefore tries to find $\pi^*$, s.t. $\mathcal{R}^{\pi^*} \geq \mathcal{R}^\pi$ for all $\pi$. If the model of the environment is unknown, model-free methods based on the Bellman Optimality Equation over the so-called action-value can be used:

$$Q^\pi(s_t, a_t) = \mathbf{E}_{a_{j > t} \sim \pi, s_{j > t} \sim \mathcal{M}} \left[ \sum_{j=t}^{T-1} \gamma^{j-t} r_j \,\middle|\, s_t, a_t \right]. \tag{2}$$

In the following, we abbreviate $\mathbf{E}_{a_{j > t} \sim \pi, s_{j > t} \sim \mathcal{M}}[\cdot | s_t, a_t]$ by $\mathbf{E}_{t,\pi,\mathcal{M}}[\cdot]$. One popular representative of continuous model-free reinforcement learning is the *Deep Deterministic Policy Gradient* algorithm (DDPG) (Lillicrap et al., 2016). In DDPG, actor $\mu$ is a deterministic mapping from states to actions, $\mu : \mathcal{S} \mapsto \mathcal{A}$, representing the actions that maximize the critic $Q^\mu$, i.e.

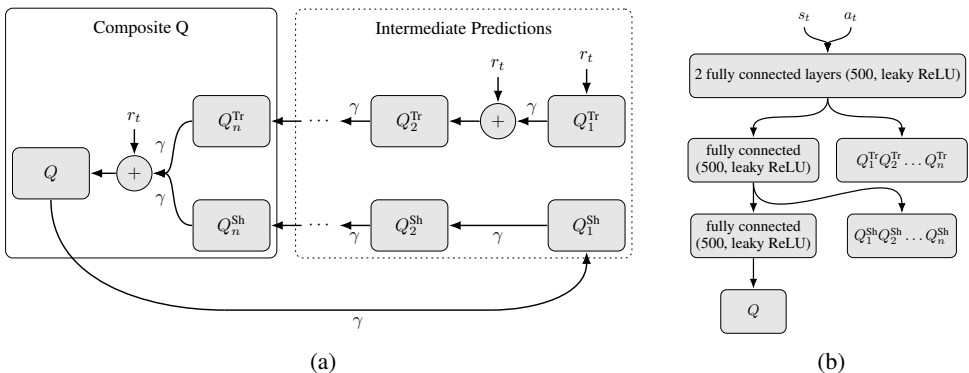

Figure 1: (a) Structure of Composite Q-learning. Target networks are omitted for visibility. $Q_i^{\cdot}$ denotes the Truncated and Shifted Q-functions at step $i$ and $Q$ the Composite Q-function. Incoming edges yield the targets for the corresponding heads. Edges denoted by $\gamma$ are discounted. (b) Architecture of the Composite Q-network used in our experiments in Section 5.2.

$\mu(s_t) = \arg\max_a Q^\mu(s_t, a)$. $Q$ and $\mu$ are estimated by function approximators $Q(\cdot, \cdot | \theta^Q)$ and $\mu(\cdot | \theta^\mu)$, parameterized by $\theta^Q$ and $\theta^\mu$. The critic is optimized on the mean squared error between predictions $Q(s_j, a_j | \theta^Q)$ and targets $y_j = r_j + \gamma Q'(s_{j+1}, \mu'(s_{j+1} | \theta^{\mu'}) | \theta^{Q'})$, where $Q'$ and $\mu'$ are target networks, parameterized by $\theta^{Q'}$ and $\theta^{\mu'}$. The parameters of $\mu$ are optimized following the deterministic policy gradient theorem (Silver et al., 2014):

$$\nabla_{\theta^\mu} \leftarrow \frac{1}{m} \sum_j \nabla_a Q(s, a | \theta^Q)|_{s=s_j, a=\mu(s_j | \theta^\mu)} \nabla_{\theta^\mu} \mu(s | \theta^\mu), \tag{3}$$

and the parameters of the target networks are updated according to:

$$\theta^{Q'} \leftarrow (1 - \tau)\theta^{Q'} + \tau\theta^Q \text{ and } \theta^{\mu'} \leftarrow (1 - \tau)\theta^{\mu'} + \tau\theta^\mu, \tag{4}$$

with $\tau \in [0, 1]$.

TD3 (Fujimoto et al., 2018) adds three adjustments to vanilla DDPG. First, the minimum prediction of two distinct critics is taken for target calculation to alleviate overestimation bias, an approach belonging to the family of Double Q-learning algorithms (van Hasselt et al., 2016). Second, Gaussian smoothing is applied to the target-policy, addressing the variance in updates. Third, actor and target networks are updated every $d$-th gradient step of the critic, to account for the problem of moving targets.

## 4 OFF-POLICY MULTI-STEP Q-LEARNING

In this section, we introduce the Composite Q-learning algorithm, an off-policy multi-step reinforcement learning method to enhance data-efficiency, along with MVE-TD3 and TD3($\Delta$) as baselines.

### 4.1 COMPOSITE Q-LEARNING

The main motivation behind this work is the assumption that learning values on short time scales can be achieved faster than for the full temporal horizon of a task, which can be prohibitively long. As Murphy (2005) shows for the fixed-batch fitted Q-iteration case, the number of samples needed to achieve a certain generalization error is exponential in the horizon of the MDP. As discussed in Jin et al. (2018), Q-learning with UCB-exploration has total regret polynomial in the horizon of the MDP and exponential with $\epsilon$-greedy exploration (Kearns and Singh, 2002). We thus argue that truncated horizons for a fixed MDP translate to lower sample complexity of value-estimation. Building upon this idea, we estimate the return of $n$-step rollouts of the target-policy via Truncated Q-functions which we then combine to the full return with model-free Shifted Q-functions, an approach we call *Composite Q-learning*, while remaining *purely off-policy*. Since these quantities cannot be estimated directly from single-step transitions, we introduce a consecutive bootstrapping

scheme based on intermediate predictions. For an overview, see Figure 1a. The full algorithm is in the appendix. Code based on the implementation of TD3[1] can be found in the supplementary.

### 4.1.1 TRUNCATED Q-FUNCTIONS

In order to formalize the off-policy estimation of $n$-step returns, assume that $n \ll (T-1)$ and that $(T - 1 - t) \mod n = 0$ for task horizon $T$. We make use of the following observation:

$$
\begin{aligned}
Q^\pi(s_t, a_t) &= \mathbf{E}_{t,\pi,\mathcal{M}} \left[ r_t + \gamma r_{t+1} + \gamma^2 r_{t+2} + \gamma^3 r_{t+3} + \cdots + \gamma^{T-1} r_{T-1} \right] \\
&= \mathbf{E}_{t,\pi,\mathcal{M}} \left[ \left( \sum_{j=t}^{t+n-1} \gamma^{j-t} r_j \right) + \gamma^n \left( \sum_{j=t+n}^{t+2n-1} \gamma^{j-(t+n)} r_j \right) \right. \\
&\qquad \left. + \cdots + \gamma^{T-n} \left( \sum_{j=T-n}^{T-1} \gamma^{j-(T-n)} r_j \right) \right].
\end{aligned}
\tag{5}
$$

That is, we can define the action-value as the combination of partial sums of length $n$. We can then define the Truncated Q-function as $Q_n^\pi(s_t, a_t) = \mathbf{E}_{t,\pi,\mathcal{M}}[\sum_{j=t}^{t+n-1} \gamma^{j-t} r_j]$, which we plug into Equation (5):

$$
Q^\pi(s_t, a_t) = \mathbf{E}_{t,\pi,\mathcal{M}}[Q_n^\pi(s_t, a_t) + \gamma^n Q_n^\pi(s_{t+n}, a_{t+n}) + \cdots + \gamma^{T-n} Q_n^\pi(s_{T-n}, a_{T-n})]. \tag{6}
$$

**Theorem 1.** *Let $Q_1^\pi(s_t, a_t) = r_t$ be the one-step Truncated Q-function and $Q_{i>1}^\pi(s_t, a_t) = r_t + \gamma \mathbf{E}_{t,\pi,\mathcal{M}}[Q_{i-1}^\pi(s_{t+1}, a_{t+1})]$ the i-step Truncated Q-function. Then $Q_i^\pi(s_t, a_t)$ represents the truncated return $Q_i^\pi(s_t, a_t) = \mathbf{E}_{t,\pi,\mathcal{M}}[\sum_{j=t}^{t+i-1} \gamma^{j-t} r_j]$.*

Following Theorem 1 (the proof can be found in the supplementary), we approximate $Q_n^\pi(s_t, a_t)$ off-policy via consecutive bootstrapping. Let $Q^{\mathrm{Tr}}(\cdot, \cdot | \theta^{Q^{\mathrm{Tr}}})$ denote a function approximator with parameters $\theta^{Q^{\mathrm{Tr}}}$ and $n$ outputs, subsequently called *heads*, estimating $Q_i^\pi$. Each output $Q_i^{\mathrm{Tr}}$ bootstraps from the prediction of the preceding head, with the first approximating the immediate reward function. The targets are therefore given by:

$$
y_{j,1}^{\mathrm{Tr}} = r_j \text{ and } y_{j,i>1}^{\mathrm{Tr}} = r_j + \gamma Q_{i-1}^{\mathrm{Tr}'}(s_{j+1}, \mu'(s_{j+1}|\theta^{\mu'})|\theta^{Q_{i-1}^{\mathrm{Tr}'}}), \tag{7}
$$

where $\mu'$ corresponds to the actor maximizing the full Q-value as defined in Section 3. That is, $Q^{\mathrm{Tr}}$ represents evaluations of $\mu$ at different stages of truncation and $y_{j,i<n}^{\mathrm{Tr}}$ serve as intermediate predictions to get $y_{j,n}^{\mathrm{Tr}}$. We then *only* use $Q_n^{\mathrm{Tr}}$, which implements the full $n$-step return, as the first part of the composition of the Q-target. Please note that in order to estimate Equation (6), the dynamics model would be needed to get $s_{t+c\cdot n}$ of a rollout starting in $s_t$. In the following, we describe an approach to achieve an estimation of Equation (6) model-free.

### 4.1.2 SHIFTED Q-FUNCTIONS

To get an estimation for the remainder of the rollout $Q_{n:\infty}^\pi = \mathbf{E}_{t,\pi,\mathcal{M}}[\gamma^n Q(s_{t+n}, a_{t+n})]$ after $n$ steps, we use a consecutive bootstrapping formulation of the Q-prediction.

**Theorem 2.** *Let $Q_{1:\infty}^\pi(s_t, a_t) = \mathbf{E}_{t,\pi,\mathcal{M}}[\gamma Q^\pi(s_{t+1}, a_{t+1})]$ be the one-step Shifted Q-function and $Q_{i>1:\infty}^\pi(s_t, a_t) = \mathbf{E}_{t,\pi,\mathcal{M}}[\gamma Q_{i-1:\infty}^\pi(s_{t+1}, a_{t+1})]$ the i-step Shifted Q-function. Then $Q_{i:\infty}^\pi(s_t, a_t)$ represents the shifted return $Q_{i:\infty}^\pi(s_t, a_t) = \mathbf{E}_{t,\pi,\mathcal{M}}[\gamma^i Q^\pi(s_{t+i}, a_{t+i})]$.*

Again, the proof of Theorem 2 can be found in the supplementary. Let $Q^{\mathrm{Sh}}(\cdot, \cdot | \theta^{Q^{\mathrm{Sh}}})$ denote the function approximator estimating the Shifted Q-function $Q_{n:\infty}^\pi$, parameterized by $\theta^{Q^{\mathrm{Sh}}}$. We can shift the Q-prediction by bootstrapping without taking the immediate reward into account, so as to skip the first $n$ rewards of a target-policy rollout. The Shifted Q-targets for heads $Q_i^{\mathrm{Sh}}$ therefore become:

$$
y_{j,1}^{\mathrm{Sh}} = \gamma Q'(s_{j+1}, \mu'(s_{j+1}|\theta^{\mu'})|\theta^{Q'}) \text{ and } y_{j,i>1}^{\mathrm{Sh}} = \gamma Q_{i-1}^{\mathrm{Sh}'}(s_{j+1}, \mu'(s_{j+1}|\theta^{\mu'})|\theta^{Q_{i-1}^{\mathrm{Sh}'}}). \tag{8}
$$

---

[1] https://github.com/sfujim/TD3

### 4.1.3 COMPOSITION

Following the definitions of Truncated and Shifted Q-functions, we can compose the full return.

**Theorem 3.** *Let* $Q_n^\pi(s_t, a_t) = \mathbf{E}_{t,\pi,\mathcal{M}}[\sum_{j=t}^{t+n-1} \gamma^{j-t} r_j]$ *be the truncated return and* $Q_{n:\infty}^\pi(s_t, a_t) = \mathbf{E}_{t,\pi,\mathcal{M}}[\gamma^n Q(s_{t+n}, a_{t+n})]$ *the shifted return. Then* $Q^\pi(s_t, a_t) = Q_n^\pi(s_t, a_t) + Q_{n:\infty}^\pi(s_t, a_t)$ *represents the full return, i.e.* $Q^\pi(s_t, a_t) = \mathbf{E}_{t,\pi,\mathcal{M}}[\sum_{j=t}^{\infty} \gamma^{j-t} r_j]$.

The incorporation of truncated returns breaks down the time scale of the long-term prediction by the Shifted Q-function. For details, see the proof of Theorem 3 in the supplementary. We can thus define the Composite Q-target as:

$$y_j^Q = r_j + \gamma(Q_n^{\mathrm{Tr}\prime}(s_{j+1}, \mu'(s_{j+1}|\theta^{\mu'})|\theta^{Q_n^{\mathrm{Tr}\prime}}) + Q_n^{\mathrm{Sh}\prime}(s_{j+1}, \mu'(s_{j+1}|\theta^{\mu'})|\theta^{Q_n^{\mathrm{Sh}\prime}})), \tag{9}$$

approximated by $Q(\cdot, \cdot|\theta^Q)$ with parameters $\theta^Q$. Since we have true reward $r_j$, we include it in the target. Please note, that shifting the action-value in time imposes a bottleneck, since it relies on the estimation of the full action-value. We therefore jointly estimate $Q^{\mathrm{Tr}}$, $Q^{\mathrm{Sh}}$ and $Q$ with function approximator $Q^{\mathcal{C}}(\cdot, \cdot|\theta^{Q^{\mathcal{C}}})$. Let $\mathrm{SE}_{\mathcal{C}}$ denote the squared error between targets $y_j^{\mathcal{C}}$ and predictions $Q^{\mathcal{C}}(s_j, a_j|\theta^{Q^{\mathcal{C}}})$. See Figure 1b for the detailed architecture of the Composite Q-network.

Each pair $Q_i^{\mathrm{Tr}} + Q_i^{\mathrm{Sh}}|_{1 \le i \le n}$ is a complete approximation of the true Q-value. The circular dependency can lead to stability issues, due to the amplification of propagated errors. We add a regularization term to the loss penalizing the deviation between the prediction of $Q$ and the $n$ different Q-pairs to keep estimates in a narrow range. It is implemented as the weighted mean squared error, i.e. the loss function becomes:

$$\mathcal{L} = \frac{1}{m} \sum_{j=1}^{m} \left( \mathrm{SE}_{\mathcal{C}} + \beta \frac{1}{n} \sum_{i=1}^{n} \left( Q(s_j, a_j|\theta^Q) - \left( Q_i^{\mathrm{Tr}}(s_j, a_j|\theta^{Q_i^{\mathrm{Tr}}}) + Q_i^{\mathrm{Sh}}(s_j, a_j|\theta^{Q_i^{\mathrm{Sh}}}) \right) \right)^2 \right), \tag{10}$$

for batch of size $m$ and with $\beta$ being the regularization weight. Actor $\mu$ is then updated on $Q$.

### 4.2 TD3($\Delta$) AND MVE-TD3

Most related to Composite Q-learning are TD($\Delta$) (Romoff et al., 2019) and MVE-DDPG (Feinberg et al., 2018). In this section, we describe how to extend TD($\Delta$) to an off-policy setting and how to combine MVE and TD3.

**TD3($\Delta$)**  Another way to divide the value function into multiple time scales is TD($\Delta$) (Romoff et al., 2019). To this point, it has only been applied in an on-policy setting. In favor of comparability, we extend TD($\Delta$) to Q-learning, yielding TD3($\Delta$). The main idea of TD($\Delta$) is the combination of different value functions corresponding to increasing discount values. Let $\gamma^\Delta$ denote a fixed ordered sequence of increasing discount values, i.e. $\gamma^\Delta = (\gamma_1, \gamma_2, \ldots, \gamma_k)^\top|_{\gamma_i > \gamma_{i-1}}$. We can then define delta functions $W_i$ as:

$$W_1 = Q_{\gamma_1} \text{ and } W_{i>1} = Q_{\gamma_i} - Q_{\gamma_{i-1}}. \tag{11}$$

Let $Q^\Delta(\cdot, \cdot|\theta^{Q^\Delta})$ denote the function approximator estimating $Q_{\gamma_{1 \le i \le k}}$. Based on the derivations in (Romoff et al., 2019), the targets for Q-learning can be formalized as:

$$y_{j,1}^\gamma = r_j + \gamma_1 Q'_{\gamma_1}(s_{j+1}, \mu'(s_{j+1}|\theta^{\mu'})|\theta^{Q'_{\gamma_1}}) \text{ and}$$

$$y_{j,i>1}^\gamma = (\gamma_i - \gamma_{i-1})Q'_{\gamma_{i-1}}(s_{j+1}, \mu'(s_{j+1}|\theta^{\mu'})|\theta^{Q'_{\gamma_{i-1}}}) + \gamma_i W'_i(s_{j+1}, \mu'(s_{j+1}|\theta^{\mu'})|\theta^{W'_i}), \tag{12}$$

which can then be used in any Q-learning algorithm. The authors suggest the use of $n$-step targets within TD($\Delta$) which is not easily applicable in an off-policy setting. In our experiments, we therefore compare our approach to single-step TD3($\Delta$). The algorithm can be found in the appendix.

**MVE-TD3**  We also apply Model-based Value Expansion within TD3, subsequently called MVE-TD3. We add Gaussian policy smoothing to the rollout of the model. In contrast to Feinberg et al. (2018), however, we do not assume to have knowledge about the reward function. Our model therefore approximates both, dynamics and reward.

## 5 EXPERIMENTAL RESULTS

We evaluate Composite Q-learning in both, the tabular setting and the actor-critic method TD3.

### 5.1 TABULAR COMPOSITE Q-LEARNING

To analyze the effect of incorporating short-term prediction $Q_n^{\text{Tr}}$ in the Q-update, we apply Composite Q-learning in the tabular case to the MDP of horizon $K$ given in Figure 2a. We compare it to vanilla Q-learning, as well as multi-step Q-learning based on subtrajectories of the exploratory policy and imaginary rollouts of the target-policy with the true model of the MDP.

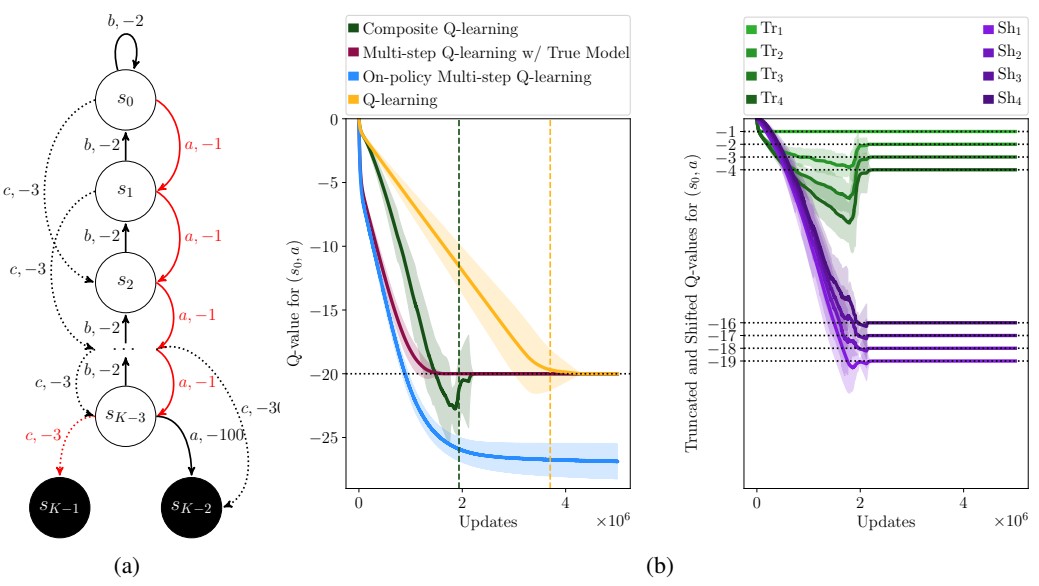

(a)                                      (b)

Figure 2: (a) In this MDP of horizon $K$, the agent ought to arrive at terminal state $s_{K-1}$ using actions $\{a, b, c\}$. The initial state is $s_0$ and the optimal policy is given in red. (b) Mean results and two standard deviations over 10 runs on the MDP with a horizon of $K = 20$. The left plot depicts the value of $s_0$ and action $a$ as estimated by the different approaches over time. Dashed lines indicate convergence to the optimal policy. The predicted Truncated Q-values for state $s_0$ and action $a$ with horizons 1 to 4, denoted by $\text{Tr}_1, \ldots, \text{Tr}_4$, and predicted Shifted Q-values for state $s_0$ and action $a$, denoted by $\text{Sh}_1, \ldots, \text{Sh}_4$, are to the right. Dotted lines indicate the true optimal respective Q-values.

Results for $K = 20$ are depicted in Figure 2b. All approaches update the Q-function with a learning rate of $10^{-3}$ on the same fixed batch of $10^3$ episodes with a percentage of $10\%$ non-optimal transitions. For the multi-step approaches, we set rollout length $n = 4$. Since there is no generalization among states in the tabular setting, we update the Shifted Q-function with a learning rate of $10^{-2}$ and the Truncated Q-functions with a learning rate of $10^{-3}$. An evaluation of different learning rates for the Shifted Q-functions is depicted in Figure 3. We compare Composite Q-learning and vanilla Q-learning to Shifted Q-learning, which corresponds to Q-learning with a one-step shifted target (without approximate $n$-step returns from a Truncated Q-function). The results show that shifting the value in time alone is slowing down convergence. Precisely, Shifted Q-learning with a learning rate of $1.0$ is equivalent to vanilla Q-learning, the same holds for Composite Q-learning with a learning rate of $10^{-3}$ for the Shifted Q-function. The counterpart with different learning rates for the Truncated Q-functions, while keeping the learning rates for the full Q-estimate and Shifted Q-functions fixed, can be seen in Figure 4. While there is improvement in convergence using a larger learning rate for the Truncated Q-functions, the results show higher variance and less benefit than the Shifted Q-functions in Figure 3. The results suggest that the interplay between short- and long-term predictions yields the most benefit if the learning rate for the Shifted Q-functions can be set to a higher value. We believe this to be due to two possible reasons. The higher learning rates for the Truncated Q-functions might lead to overfitting to the shorter horizons (given that it represents an easier learning problem compared to the full return), whereas the Shifted Q-functions allow for

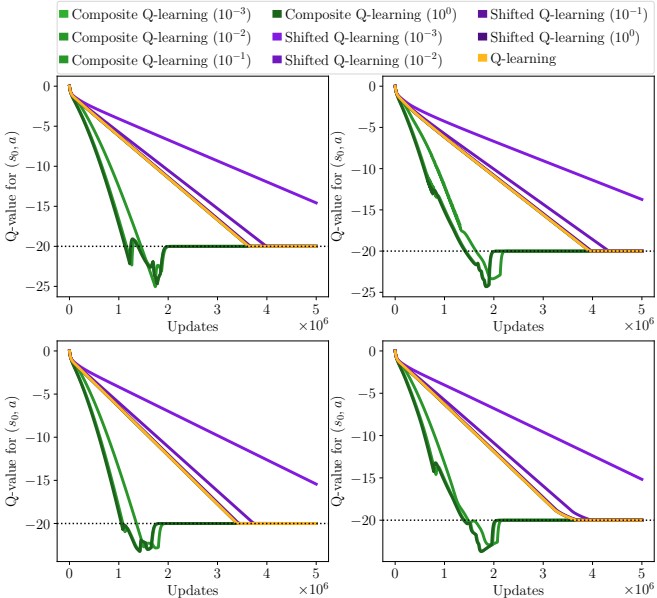

Figure 3: Results of four individual runs on the MDP with a horizon of $K = 20$ for Composite and Shifted Q-learning, with different learning rates for the Shifted Q-function (denoted by the numbers in parentheses). The learning rates for the full Q-function and for the Truncated Q-functions are set to $10^{-3}$ in all experiments.

higher learning rates, since they only have to consider the variability of the distribution over next states leading to decreased variance in their targets. In the experiments in Section 5.2, we show that this can also be achieved by generalization as in the given architecture in Figure 1b. The erroneous updates of on-policy multi-step Q-learning lead to convergence to a wrong action-value which is underlining the importance of truly off-policy learning. However, if the true model can be used for the rollout, this can be really effective. The difference in convergence speed between Q-learning and Composite Q-learning is highly significant ($p$-value of $4 \cdot 10^{-8}$ according to a $t$-test) and grows with increasing horizon, as shown in Table 1. This is in line with the findings of Jin et al. (2018) who establish a connection between the horizon of an MDP and the sample complexity of value estimation.

Table 1: Comparison of convergence speed between tabular Q-learning and tabular Composite Q-learning for exemplary runs on the MDP given in Figure 2a with $n = 4$.

| Horizon $K$ | 10 | 20 | 50 | 100 |
|---|---|---|---|---|
| Speed up to Q-learning | 11% | 44% | 57% | 66% |

## 5.2 Composite Q-learning with Function Approximation

We evaluate TD3, Composite TD3, MVE-TD3 and TD3($\Delta$) on three robot simulation tasks of OpenAI Gym (Brockman et al., 2016) based on MuJoCo (Todorov et al., 2012): Walker2d-v2, Ant-v2 and Hopper-v2. A visualization of the environments is depicted in Figure 5.

**Parameter Setting** Our main focus is on the analysis of the structure of Q-functions and not to achieve maximum possible performance. We therefore keep the main parameters the same across all approaches (the underlying algorithm was TD3 in all cases), since this would lead to another source of potential differences otherwise. Learning rate ($10^{-3}$), target update ($5 \cdot 10^{-3}$) and actor setting (two hidden layers with 400 and 300 neurons and ReLU activation) are the same as in the default setting of TD3. We use Gaussian exploration noise with $\sigma = 0.15$. The critic in Composite Q-learning consists of four layers with 500 neurons and leaky ReLU activation, the critic in all other

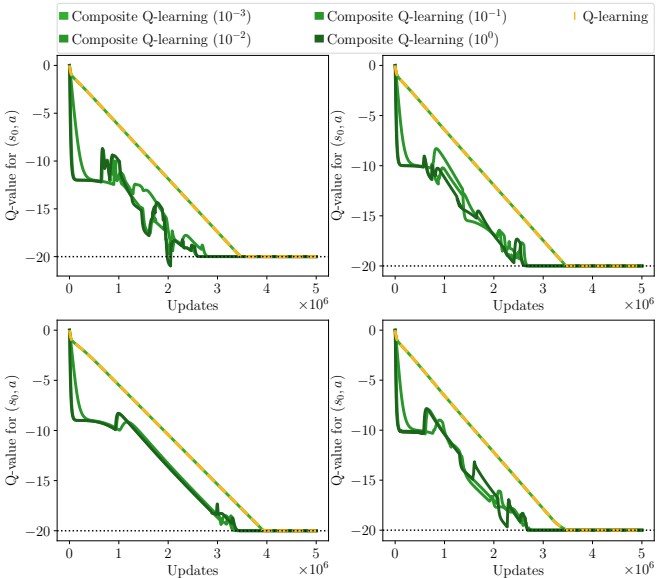

Figure 4: Results of four individual runs on the MDP with a horizon of $K = 20$ for Composite Q-learning, with different learning rates for the Truncated Q-function (denoted by the numbers in parentheses). The learning rates for the full Q-function and for the Shifted Q-functions are set to $10^{-3}$ in all experiments.

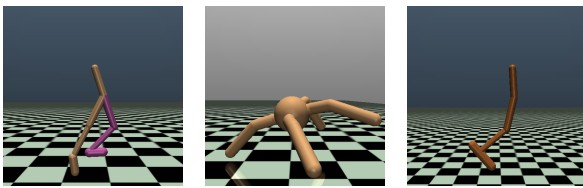

Figure 5: Visualization of Walker2d-v2 (left), Ant-v2 (middle) and Hopper-v2 (right).

approaches and the model in MVE-TD3, with a learning rate of $10^{-3}$, has two layers. However, we also evaluate TD3 and TD3($\Delta$) with the same critic setting as Composite TD3, see Figure 8 in the supplementary. This corresponds to the most stable setting we could find for TD3 with the given subset of MuJoCo tasks. For Composite TD3, we set $n = 50$ and $\beta = 10^{-4}$ for Walker2d-v2 and Hopper-v2 and $\beta = 5 \cdot 10^{-5}$ for Ant-v2. For TD3($\Delta$), we use the $\gamma$-schedule as suggested by the authors, i.e. $\gamma_1 = 0$ and $\gamma_{i>1} = \frac{\gamma_{i-1}+1}{2}$, with an upper limit of 0.99 (Romoff et al., 2019). For MVE-TD3, we use a rollout length of 3, as described in Feinberg et al. (2018).

**Data Efficiency** We choose the area under the learning curve as performance measure, which is a common way to evaluate data-efficiency and learning stability (see e.g. Hessel et al. (2018)). Since vanilla TD3 can have severe stability issues for a small number of runs, we compare median and interquartile ranges (IQR). As listed in Table 2, Composite TD3 outperforms TD3, as well as state-of-the-art multi-step approaches, in terms of learning speed throughout training. After the considered time frame of $4 \cdot 10^5$ transitions, Composite TD3 has a 19%, 11% and 19% larger area under the median learning curve, in comparison to vanilla TD3. As depicted in Figure 6, MVE-TD3 is highly sensitive w.r.t. accumulating errors in reward and state prediction, even when applying the TD-$k$ trick. However, if dynamics and reward are estimated with high accuracy, model-based rollouts can be very effective, as the results show for Ant-v2. Based on our experiments, TD3($\Delta$) seems to be less sensitive than MVE-TD3, albeit also less efficient than Composite TD3. The maximum return found by all approaches within $4 \cdot 10^5$ transitions can be seen in Table 3.

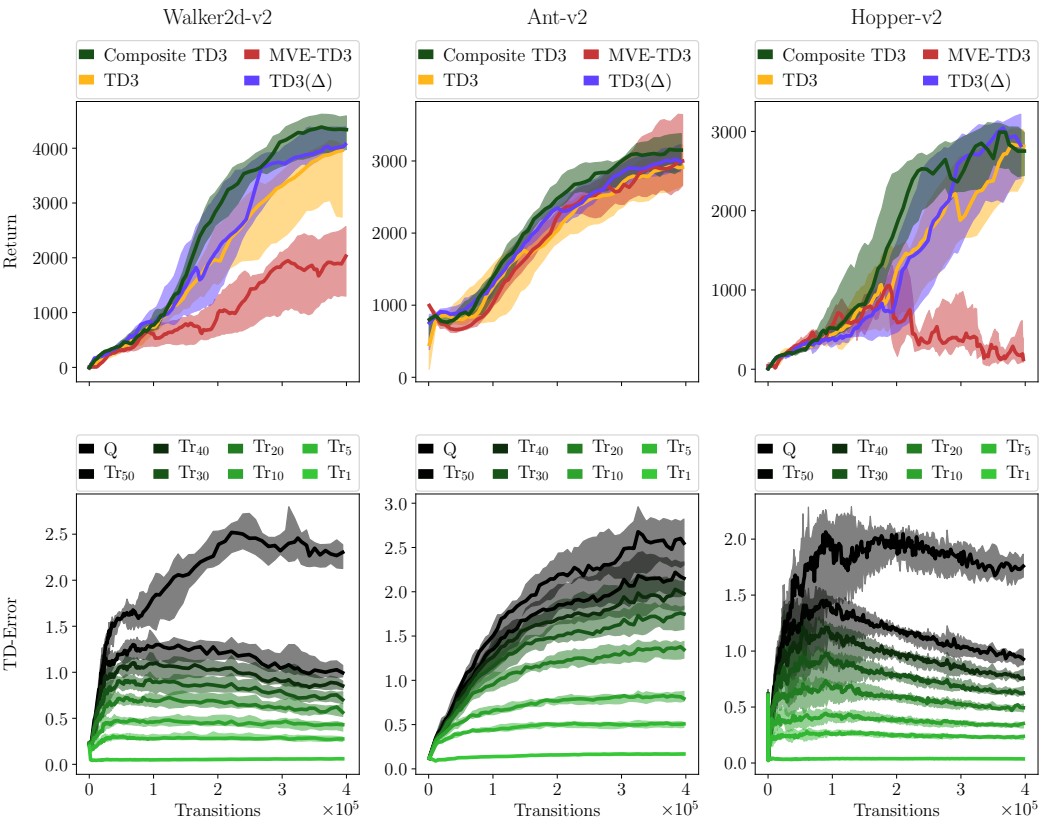

Figure 6: Results (top) and TD-errors (bottom) for Walker2d-v2 (left), Ant-v2 (middle) and Hopper-v2 (right). The plots show median and interquartile ranges over 11 training runs, each representing mean evaluation performance over 100 initial states. The lower plots show TD-errors over time for the different horizons of the Truncated Q-function (denoted by $\text{Tr}_h$ for horizon $h$) as well as the TD-errors for the complete Q-estimate. Please note that *TD-error* here means the deviation from the associated target.

Table 2: Normalized area under the median learning curve and IQRs over 11 training runs.

| Samples | Method | Walker2d-v2 | Ant-v2 | Hopper-v2 |
|---|---|---|---|---|
| $2 \cdot 10^5$ | TD3 | 82% (-19, +15) | 88% (-20, +19) | 84% (-29, +28) |
| | Composite TD3 | **100% (-11, +15)** | **100% (-11, +8)** | **100% (-20, +50)** |
| | MVE-TD3 | 50% (-15, +14) | 82% (-5, +10) | 86% (-17, +20) |
| | TD3($\Delta$) | 89% (-26, +34) | 97% (-6, +8) | 74% (-22, +24) |
| $3 \cdot 10^5$ | TD3 | 79% (-19, +14) | 88% (-16, +13) | 78% (-28, +21) |
| | Composite TD3 | **100% (-10, +12)** | **100% (-9, +8)** | **100% (-19, +34)** |
| | MVE-TD3 | 44% (-16, +12) | 86% (-6, +8) | 57% (-13, +18) |
| | TD3($\Delta$) | 87% (-21, +28) | 94% (-6, +8) | 72% (-26, +27) |
| $4 \cdot 10^5$ | TD3 | 81% (-20, +11) | 89% (-14, +11) | 81% (-25, +19) |
| | Composite TD3 | **100% (-10, +10)** | **100% (-9, +7)** | **100% (-18, +27)** |
| | MVE-TD3 | 44% (-16, +12) | 88% (-8, +11) | 44% (-11, +16) |
| | TD3($\Delta$) | 88% (-16, +22) | 95% (-6, +7) | 80% (-27, +22) |

**TD-error Analysis** To test our intuition stated in the motivation, we refer to the lower row of Figure 6 which shows the TD-errors for the different stages of truncation, as well as the TD-errors for the complete Q-estimate. All Truncated Q-estimates have lower TD-error throughout learning and it can be seen that the TD-error is consistently higher for longer horizons. This reaffirms our

Table 3: Maximum return of the median learning curve and IQRs over 11 training runs within the considered time frame of $4 \cdot 10^5$ samples.

| Environment | TD3 | Composite TD3 | MVE-TD3 | TD3($\Delta$) |
|---|---|---|---|---|
| Walker2d-v2 | 3962 (-887, +187) | **4391 (-414, +239)** | 2050 (-703, +524) | 4101 (-70, +304) |
| Ant-v2 | 2933 (-229, +291) | **3172 (-255, +250)** | 3019 (-322, +625) | 3043 (-171, +199) |
| Hopper-v2 | 2888 (-460, +169) | 3044 (-459, +103) | 1081 (-244, +234) | **3047 (-474, +172)** |

hypothesis that learning is easier on shorter horizons. The Composite Q-target therefore reflects the true action-value faster which is beneficial for Q-learning.

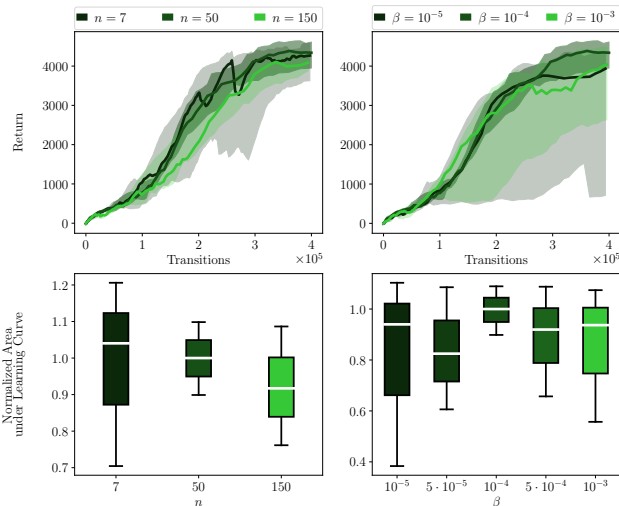

Figure 7: Results (top) and normalized area under the learning curve (bottom) for Composite TD3 in the Walker2d-v2 environment with different truncation horizons $n$ (left) and different regularization weights $\beta$ (right). The plots show median and interquartile ranges over 11 training runs, each representing mean evaluation performance over 100 initial states.

**Sensitivity to Hyperparameters** Lastly, we evaluate the influence of $n$ and $\beta$, exemplary for the Walker2d-v2 environment. The results can be seen in Figure 7. Shorter truncation horizons can lead to faster convergence, but the variance increases. If $n$ is rather large, however, it can slow learning down. The same holds if the regularization weight $\beta$ is set to a high value. On the other hand, regularization is needed to keep the predictions in a narrow range, since it can lead to stability issues, otherwise.

## 6 CONCLUSION

We introduced Composite Q-learning, an off-policy learning method that divides the long-term value into smaller time scales. It combines Truncated Q-functions acting on a short horizon with Shifted Q-functions for the remainder of the rollout. We analyzed the efficacy of Composite Q-learning in the tabular case and showed that the benefit of short-term predictions increases with growing task horizon. We further evaluated MVE-TD3 and introduced TD3($\Delta$), an off-policy variant of TD($\Delta$). We showed on three simulated robot tasks that Composite TD3 outperforms vanilla TD3 by 19%, 11% and 19% in terms of area under the median learning curve. The given results provide evidence that Composite TD3 enhances data-efficiency compared to other approaches in off-policy multi-step learning.

Going forward, the uncertainty estimate based on the variance of the Composite Q-network could be of benefit in both, update calculation and exploration. We further leave the application of the truncated formulation of other quantities such as state change or auxiliary costs as future work.

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

## A  ALGORITHM FOR COMPOSITE TD3

A detailed description of Composite DDPG is given in Algorithm 1, where the adjustments of TD3 are omitted for simplicity. In order to transform Algorithm 1 to its TD3-equivalent, Gaussian policy smoothing has to be added to all targets in Line 8, as well as taking the minimum prediction of two distinct critics for each target. Furthermore, actor and target networks have to be updated with delay.

---

**Algorithm 1:** Composite DDPG

---

1  initialize critic $Q^{\mathcal{C}}$, actor $\mu$ and targets $Q^{\mathcal{C}\prime}$, $\mu'$
2  initialize replay buffer $\mathcal{R}$
3  **for** $episode = 1..E$ **do**
4      get initial state $s_1$
5      **for** $t = 1..T$ **do**
6          apply action $a_t = \mu(s_t|\theta^\mu) + \xi$, where $\xi \sim \mathcal{N}(0, \sigma)$
7          observe $s_{t+1}$ and $r_t$ and save transition $(s_t, a_t, s_{t+1}, r_t)$ in $\mathcal{R}$
8          calculate targets:

$$y_{j,1}^{\mathrm{Tr}} = r_j$$
$$y_{j,i>1}^{\mathrm{Tr}} = r_j + \gamma Q_{i-1}^{\mathrm{Tr}\prime}(s_{j+1}, \mu'(s_{j+1}|\theta^{\mu'})|\theta^{Q_{i-1}^{\mathrm{Tr}\prime}})$$
$$y_{j,1}^{\mathrm{Sh}} = \gamma Q'(s_{j+1}, \mu'(s_{j+1}|\theta^{\mu'})|\theta^{Q'})$$
$$y_{j,i>1}^{\mathrm{Sh}} = \gamma Q_{i-1}^{\mathrm{Sh}\prime}(s_{j+1}, \mu'(s_{j+1}|\theta^{\mu'})|\theta^{Q_{i-1}^{\mathrm{Sh}\prime}})$$
$$y_j^Q = r_j + \gamma(Q_n^{\mathrm{Tr}\prime}(s_{j+1}, \mu'(s_{j+1}|\theta^{\mu'})|\theta^{Q_n^{\mathrm{Tr}\prime}}) + Q_n^{\mathrm{Sh}\prime}(s_{j+1}, \mu'(s_{j+1}|\theta^{\mu'})|\theta^{Q_n^{\mathrm{Sh}\prime}}))$$
$$y_j^{\mathcal{C}} = \left[ y_j^Q, y_{j,1}^{\mathrm{Tr}}, y_{j,i>1}^{\mathrm{Tr}}, y_{j,1}^{\mathrm{Sh}}, y_{j,i>1}^{\mathrm{Sh}} \right]$$

9          update $Q^{\mathcal{C}}$ on minibatch $b$ of size $m$ from $\mathcal{R}$ according to Equation (10)
10          update $\mu$ on $Q$
11          adjust parameters of $Q^{\mathcal{C}\prime}$ and $\mu'$

---

## B  ALGORITHM FOR TD3($\Delta$)

A detailed description of DDPG($\Delta$) is given in Algorithm 2. The authors suggest the use of $n$-step samples which is not easily applicable in an off-policy setting. In our experiments, we therefore compare our approach to single-step TD3($\Delta$). To transform DDPG($\Delta$) to TD3($\Delta$), the adjustments as described in Appendix A have to be applied analogously.

---

**Algorithm 2:** DDPG($\Delta$)

---

1  initialize critic $Q^{\Delta}$, actor $\mu$ and targets $Q^{\Delta\prime}$, $\mu'$
2  initialize replay buffer $\mathcal{R}$
3  set discount values $\gamma^{\Delta} = (\gamma_0, \gamma_1, \ldots, \gamma_k)^{\intercal}$
4  **for** $episode = 1..E$ **do**
5      get initial state $s_1$
6      **for** $t = 1..T$ **do**
7          apply action $a_t = \mu(s_t|\theta^\mu) + \xi$, where $\xi \sim \mathcal{N}(0, \sigma)$
8          observe $s_{t+1}$ and $r_t$ and save transition $(s_t, a_t, s_{t+1}, r_t)$ in $\mathcal{R}$
9          calculate targets:

$$y_{j,1}^{\gamma} = r_j + \gamma_1 Q_{\gamma_1}'(s_{j+1}, \mu'(s_{j+1}|\theta^{\mu'})|\theta^{Q_{\gamma_1}'})$$
$$y_{j,i>1}^{\gamma} = (\gamma_i - \gamma_{i-1})Q_{\gamma_{i-1}}'(s_{j+1}, \mu'(s_{j+1}|\theta^{\mu'})|\theta^{Q_{\gamma_{i-1}}'})$$
$$+ \gamma_i W_i'(s_{j+1}, \mu'(s_{j+1}|\theta^{\mu'})|\theta^{W_i'})$$

10          update $Q^{\Delta}$ on minibatch $b$ of size $m$ from $\mathcal{R}$
11          update $\mu$ on $Q_{\gamma_k}$
12          adjust parameters of $Q^{\Delta\prime}$ and $\mu'$

---

## C  TRUNCATED Q-FUNCTIONS

**Theorem 1.** *Let $Q_1^\pi(s_t, a_t) = r_t$ be the one-step Truncated Q-function and $Q_{i>1}^\pi(s_t, a_t) = r_t + \gamma \mathbf{E}_{t,\pi,\mathcal{M}}[Q_{i-1}^\pi(s_{t+1}, a_{t+1})]$ the i-step Truncated Q-function. Then $Q_i^\pi(s_t, a_t)$ represents the truncated return $Q_i^\pi(s_t, a_t) = \mathbf{E}_{t,\pi,\mathcal{M}}[\sum_{j=t}^{t+i-1} \gamma^{j-t} r_j]$.*

*Proof.* Proof by induction. $Q_1^\pi(s_t, a_t) = r_t$ by definition. The theorem follows from induction step:

$$Q_i^\pi(s_t, a_t) = r_t + \gamma \mathbf{E}_{t,\pi,\mathcal{M}} \left[ Q_{i-1}^\pi(s_{t+1}, a_{t+1}) \right]$$
$$= r_t + \gamma \mathbf{E}_{t,\pi,\mathcal{M}} \left[ \sum_{j=(t+1)}^{(t+1)+(i-1)-1} \gamma^{j-(t+1)} r_j \right]$$
$$= r_t + \gamma \mathbf{E}_{t,\pi,\mathcal{M}} \left[ \sum_{j=(t+1)}^{t+i-1} \gamma^{j-(t+1)} r_j \right]$$
$$= r_t + \mathbf{E}_{t,\pi,\mathcal{M}} \left[ \sum_{j=(t+1)}^{t+i-1} \gamma^{j-t} r_j \right]$$
$$= \mathbf{E}_{t,\pi,\mathcal{M}} \left[ \sum_{j=t}^{t+i-1} \gamma^{j-t} r_j \right].$$

$\square$

## D  SHIFTED Q-FUNCTIONS

**Theorem 2.** *Let $Q_{1:\infty}^\pi(s_t, a_t) = \mathbf{E}_{t,\pi,\mathcal{M}}[\gamma Q^\pi(s_{t+1}, a_{t+1})]$ be the one-step Shifted Q-function and $Q_{i>1:\infty}^\pi(s_t, a_t) = \mathbf{E}_{t,\pi,\mathcal{M}}[\gamma Q_{i-1:\infty}^\pi(s_{t+1}, a_{t+1})]$ the i-step Shifted Q-function. Then $Q_{i:\infty}^\pi(s_t, a_t)$ represents the shifted return $Q_{i:\infty}^\pi(s_t, a_t) = \mathbf{E}_{t,\pi,\mathcal{M}}[\gamma^i Q^\pi(s_{t+i}, a_{t+i})]$.*

*Proof.* Proof by induction. $Q_{1:\infty}^\pi(s_t, a_t) = \mathbf{E}_{t,\pi,\mathcal{M}}[\gamma Q^\pi(s_{t+1}, a_{t+1})]$ by definition. The theorem follows from induction step:

$$Q_{i:\infty}^\pi(s_t, a_t) = \mathbf{E}_{t,\pi,\mathcal{M}} \left[ \gamma Q_{i-1:\infty}^\pi(s_{t+1}, a_{t+1}) \right]$$
$$= \mathbf{E}_{t,\pi,\mathcal{M}} \left[ \gamma(\gamma^{i-1} Q^\pi(s_{t+1+i-1}, a_{t+1+i-1})) \right]$$
$$= \mathbf{E}_{t,\pi,\mathcal{M}} \left[ \gamma(\gamma^{i-1} Q^\pi(s_{t+i}, a_{t+i})) \right]$$
$$= \mathbf{E}_{t,\pi,\mathcal{M}} \left[ \gamma^i Q^\pi(s_{t+i}, a_{t+i}) \right].$$

$\square$

## E  RESOLVING THE MUTUAL RECURSION

**Theorem 3.** *Let $Q_n^\pi(s_t, a_t) = \mathbf{E}_{t,\pi,\mathcal{M}}[\sum_{j=t}^{t+n-1} \gamma^{j-t} r_j]$ be the truncated return and $Q_{n:\infty}^\pi(s_t, a_t) = \mathbf{E}_{t,\pi,\mathcal{M}}[\gamma^n Q(s_{t+n}, a_{t+n})]$ the shifted return. Then $Q^\pi(s_t, a_t) = Q_n^\pi(s_t, a_t) + Q_{n:\infty}^\pi(s_t, a_t)$ represents the full return, i.e. $Q^\pi(s_t, a_t) = \mathbf{E}_{t,\pi,\mathcal{M}}[\sum_{j=t}^{\infty} \gamma^{j-t} r_j]$.*

*Proof.*

$$Q^\pi(s_t, a_t) = Q_n^\pi(s_t, a_t) + Q_{n:\infty}^\pi(s_t, a_t)$$

$$= \mathbf{E}_{t,\pi,\mathcal{M}} \left[ \sum_{j=t}^{t+n-1} \gamma^{j-t} r_j + \gamma^n Q^\pi(s_{t+n}, a_{t+n}) \right]$$

$$= \mathbf{E}_{t,\pi,\mathcal{M}} \left[ \sum_{j=t}^{t+n-1} \gamma^{j-t} r_j + \gamma^n \left( Q_n^\pi(s_{t+n}, a_{t+n}) + Q_{n:\infty}^\pi(s_{t+n}, a_{t+n}) \right) \right]$$

$$= \mathbf{E}_{t,\pi,\mathcal{M}} \left[ \sum_{j=t}^{t+n-1} \gamma^{j-t} r_j + \gamma^n \left( \sum_{j=t+n}^{t+2n-1} \gamma^{j-t-n} r_j + \gamma^n Q^\pi(s_{t+2n}, a_{t+2n}) \right) \right]$$

$$= \mathbf{E}_{t,\pi,\mathcal{M}} \left[ \sum_{j=t}^{t+n-1} \gamma^{j-t} r_j + \sum_{j=t+n}^{t+2n-1} \gamma^{j-t} r_j + \gamma^{2n} Q^\pi(s_{t+2n}, a_{t+2n}) \right]$$

$$= \mathbf{E}_{t,\pi,\mathcal{M}} \left[ \sum_{j=t}^{t+2n-1} \gamma^{j-t} r_j + \gamma^{2n} Q^\pi(s_{t+2n}, a_{t+2n}) \right].$$

Repeating this process then gives $Q^\pi(s_t, a_t) = \mathbf{E}_{t,\pi,\mathcal{M}}[\sum_{j=t}^{\infty} \gamma^{j-t} r_j]$. $\qquad\square$

Due to the composite structure, the Shifted Q-function represents the long-term sum of partial returns provided by the Truncated Q-function, as opposed to single reward values.

## F   TD3 AND TD3($\Delta$) WITH LARGER NETWORKS

To further analyze the impact of the Composite Q-learning structure, we evaluate TD3 and TD3($\Delta$) with the same number of parameters for the critic, as depicted in Figure 8. The approaches do not seem to make use of the additional capacity and reveal a worse performance in the given time frame.

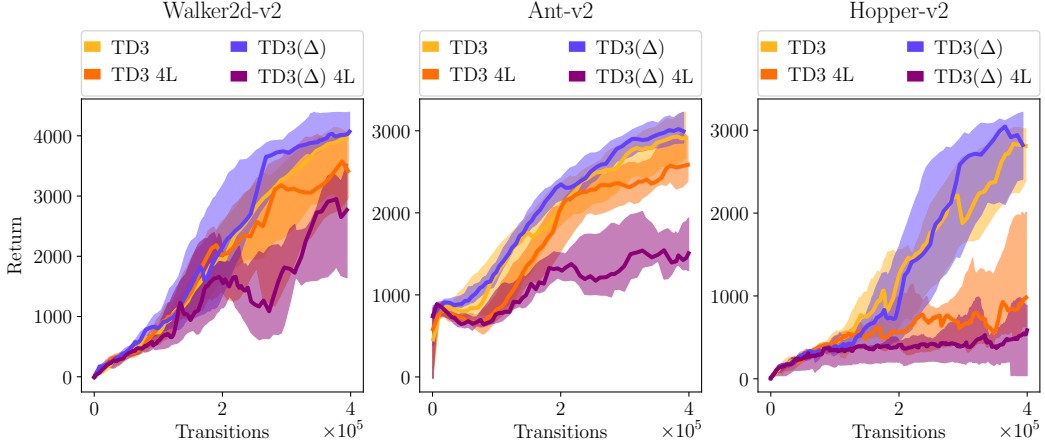

Figure 8: Results for TD3 and TD3($\Delta$) with the same number of parameters for the critic as Composite TD3, i.e. four layers with 500 neurons. The plots show median and interquartile ranges over 11 training runs, each representing mean evaluation performance over 100 initial states. We denote approaches using a critic with four layers by *4L*.

## G    SHALLOW ARCHITECTURE

In order to analyze importance of the multi-layered structure reflecting the temporal dependencies between short- and long-term predictions, we compare the architecture used in our experiments to a shallow architecture which includes all predictions as outputs of the last hidden layer for the Walker2d-v2 environment in Figure 9. Based on the results in Appendix F, we reduced the amount of hidden layers to two.

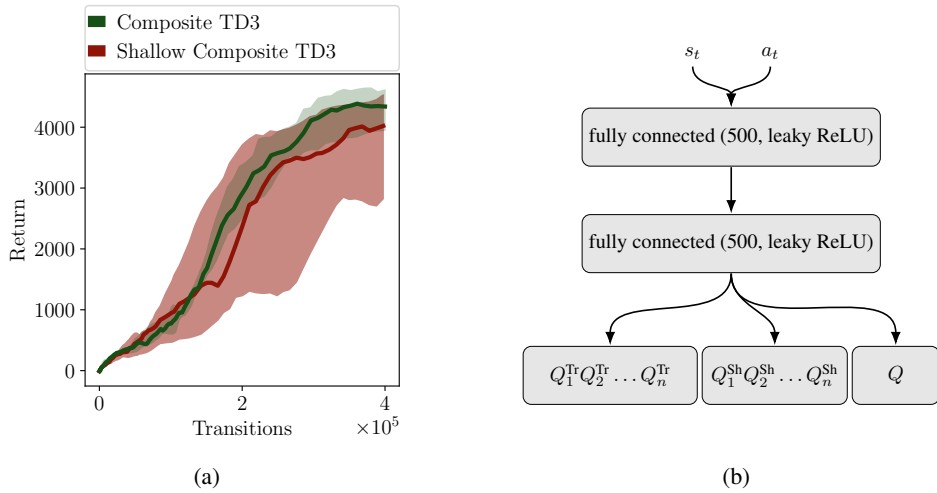

(a)                                                                (b)

Figure 9: (a) Results for a shallow architecture of the Composite Q-network for the Walker2d-v2 environment. The plots show median and interquartile ranges over 11 training runs, each representing mean evaluation performance over 100 initial states. (b) Architecture of the shallow Composite Q-network.

## H    INDIVIDUAL COMPARISONS

For clarity, we provide individual comparisons between Composite TD3 and all baselines.

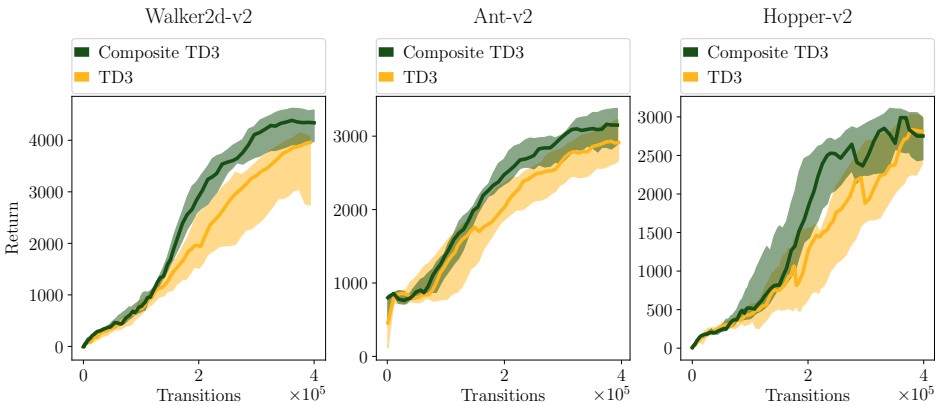

Figure 10: Results for Composite TD3 and TD3 as in Figure 6.

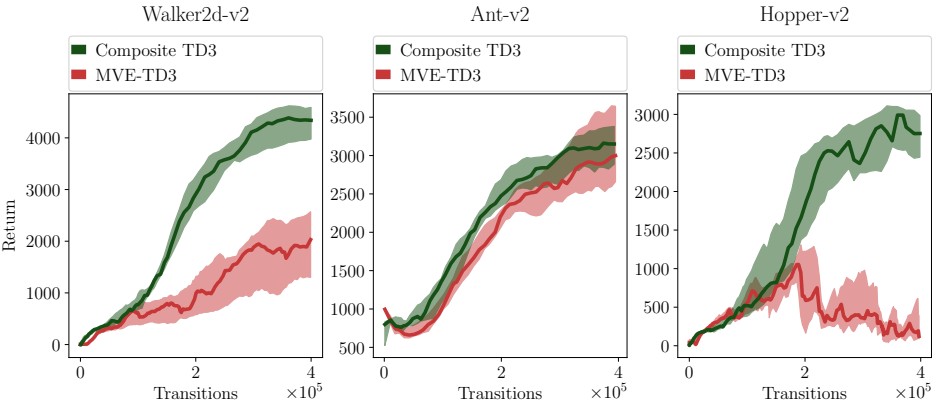

Figure 11: Results for Composite TD3 and MVE-TD3 as in Figure 6.

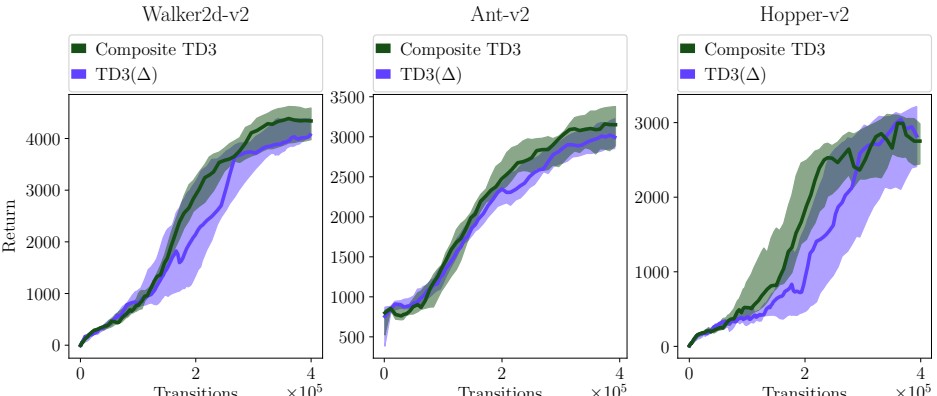

Figure 12: Results for Composite TD3 and TD3($\Delta$) as in Figure 6.

