# OpenReview forum: "Off-policy Multi-step Q-learning"
_ICLR.cc/2020/Conference — Reject_

### Official Review · AnonReviewer1 · 2019-10-21
**Official Blind Review #1**

**Rating:** 3

**Review:**

Summary

This paper introduces a new Q-learning formalism that helps reduce the bias of single step bootstrapping in Q-learning by learning multiple single step bootstrapping Q functions in parallel. This is accomplished by composing multiple n-step returns, showing that a recursive definition of n-step returns allows each return to be learned using only a single step of bootstrapping instead of at most n steps of bootstrapping. The paper solves the problem of the n-step fixed horizon by additionally composing a gamma discounted Q function that is shifted by n. In the end, the Q function used for behavior still predicts the same values as vanilla Q-learning, but with significantly less bias without a large increase in variance.

Review

I find this paper to be novel and insightful, the proposed algorithm is well supported theoretically and reasonably well supported empirically. I appreciated the careful demonstration on the smaller MDP with tabular features, showing the effects of multi-step Q-learning and clearly demonstrating the bias due to not truly using an off-policy formulation. I find that the demonstrations on the larger environments appear promising and suggest that composite multi-step Q-learning is a promising direction.

The error bars in the larger demonstrations, Figure 4, make it difficult to distinguish any meaningful differences between the algorithms. I appreciate that results are averaged over 11 runs, fortunately far more than seems to be standard at the moment, but still the amount of variance makes it difficult to say anything statistically. Table 2, then shows a reduction of the results but without mention of variance. It would be useful to include error measurements (perhaps the standard error over runs) to Table 2 to see the statistical significance of those results. Based on Figure 4, my guess is that there is negligible difference statistically.

The parameter sensitivity curves for the Walker2d domain also demonstrate that it is difficult to say anything meaningful about each parameter choice. Running a larger number of parameter settings would help to establish a clear pattern, or running each parameter setting for more independent runs could have allowed more significant results. The variance exhibited by one value in the regularization sensitivity curve is alone extremely interesting; perhaps using a different visualization that allowed more clear comparisons of the variance over independent runs would further motivate the utility of the regularization parameter. I think these results are interesting, but as presented do not sufficiently highlight the differences between the proposed algorithm and its competitors.

For the experiment in Figure 2, why not include multi-step Q-learning with importance sampling corrections on the later steps? I believe this would have fixed the bias issue, though clearly would be a tradeoff for high variance. I think this would make for a more convincing argument. Additionally, the caption does not well explain what the four green lines at the top of the plot represent. It was difficult to interpret the plot on the first pass of the paper because of this omission. Regardless, I find the results in Figure 2 to be otherwise intriguing.

Finally, the choice of meta-parameters in this paper could negatively impact results in favor of the competitor algorithms. By choosing to fix meta-parameters based on the defaults of a competitor, this could be harmfully biasing the proposed algorithm by preventing it from choosing a better stepsize. In fact, I would suspect that the proposed algorithm would exhibit lower variance updates than TD3, meaning it could potentially take advantage of higher stepsizes. This omission makes the claims of this paper weaker than they could possibly be, leaving a slight hole in the research.

---------
Edit after discussion and rebuttal phase:

I read the in-depth discussion between the authors and R3 and looked at the edits to the draft. I agree with the other reviewers on the basis of understanding the importance of meta-parameter selection. During the initial review, I found the ideas of the paper interesting enough to largely out-weigh the importance of a careful meta-parameter study. After R3's demonstration that there were indeed flaws with the results under the current meta-parameter selections, I think the best course of action would be to reject the paper in its current form.

I still strongly believe there is a place in the literature for this paper, so I hope to see this paper again at the next conference.

**Experience Assessment:**

I have published one or two papers in this area.

**Review Assessment: Checking Correctness Of Derivations And Theory:**

I assessed the sensibility of the derivations and theory.

**Review Assessment: Checking Correctness Of Experiments:**

I carefully checked the experiments.

**Review Assessment: Thoroughness In Paper Reading:**

I read the paper at least twice and used my best judgement in assessing the paper.

---

> ### Author Response · Authors · 2019-11-14
> **Response to Reviewer #1**
>
> We appreciate the constructive feedback and detailed suggestions. We included most of them in the new revision.
>
> 1) "It would be useful to include error measurements (perhaps the standard error over runs) to Table 2 to see the statistical significance of those results."
>
> We included variance measures in Table 2 and 3.
>
> 2) "Running a larger number of parameter settings would help to establish a clear pattern, or running each parameter setting for more independent runs could have allowed more significant results."
>
> We included boxplots w.r.t. the area under the learning curve to give a better visualization of the variances. We further added two more settings of the regularization weight.
>
> 3) "For the experiment in Figure 2, why not include multi-step Q-learning with importance sampling corrections on the later steps?"
>
> Within an off-policy learning regime based on deterministic policies, it is unclear how to include importance sampling in a multi-step setting. In the most naive way, the importance sampling weight can either become 0 or 1 and should be mostly 0 in the later course of learning as the target policy progresses. We therefore did not add importance sampling as a baseline here.
>
> 4) "Additionally, the caption does not well explain what the four green lines at the top of the plot represent. It was difficult to interpret the plot on the first pass of the paper because of this omission. Regardless, I find the results in Figure 2 to be otherwise intriguing."
>
> We thank the reviewer for the positive feedback and updated the submission accordingly.
>
> 5) "Finally, the choice of meta-parameters in this paper could negatively impact results in favor of the competitor algorithms."
>
> We would like to thank the reviewer for the suggestion and agree that hyperparameter optimization could be of great use here. Within the scope of the paper, however, we were not aiming at maximum performance. We wanted to analyze the influence of the structure of Q-functions within target calculation. We therefore kept crucial parameters, such as the target update and the learning rate, the same, since it would be even harder to distinguish the influence of the different methodological choices.

---

### Official Review · AnonReviewer2 · 2019-10-22
**Official Blind Review #2**

**Rating:** 3

**Review:**

  *Synopsis*:
  This paper proposes to split the value function into two separately learned components (a short-term truncated value function, and a long-term shifted value function) suggesting the short term truncated returns should learn faster as compared to the tail of the returns. They provide temporal difference formulations for a truncated value function and shifted value function, enabling efficient learning of the two components. They also provide derivations of other similar approaches to the off-policy case. Finally, they compare their algorithm to several approaches on a subset of the MuJoCo tasks, and a novel tabular domain.

  Main Contributions:
  - An algorithm, Composite Q-learning, which decomposes the value function into a short-term truncated portion and a long-term shifted portion.
  - Derivation of prior art for off-policy.

  *Review*
  The paper is generally well written (some suggestions for improved readability can be found below), and provides some nice algorithms for the community. I especially appreciate the author's willingness to derive off-policy variants of related algorithms to compared, as opposed to relegating this to future work which is the typical case. The theory for the truncated and shifted value functions also seems correct at a light check. Overall, I am recommending this paper for a weak accept as I have some concerns over the experimental results that I would like clarified. (specifically C1, Q4, and Q6).


  [Q]uestions/[C]larifications/[S]uggestions:

  C1: For the tabular domain, are the reported results over multiple runs? If not, I think it would be worthwhile to do some more runs and provide a significance test.

  C2: It would be beneficial to add some indication what the true value for state s_0 is in the plot (either with a horizontal dotted) for each of the methods (i.e. I would expect Tr0 to converge to a different value compared with composite Q-learning). Also, I'm unsure if you appropriately specified what Tr_0, Tr_1, ... are in the text. I might be missing this, but I think it should be more clear.

  S3: It might be interesting to look at the value of the shifted Q-function for this domain. Also, in the appendix I think it would be worthwhile to include the results for all of the states in the MDP (or a representative subset).

  Q4: What are the default settings for TD3 and how were they set? This is an important detail to include, even if you believe they are well accepted in the field. This will make it easier to reproduce your experiments for future work. I think it seriously harms the paper by not tuning the algorithms appropriately.

  S5: It seems as if you are using an open source implementation of TD3, if this is the case you should state this and give a link to the implementation (if you implemented yourself disregard this)

  Q6: How significant are the results in figure 4, say for Walker2d? From what I understand about IQR, significance is measured based on overlap of the medians with the competing IQRs. For example, if we look at Walker2d much of the Composite TD3 median learning curve is within the IQR of TD3(\Delta) and there are many points where TD3(\Delta) is also in the IQR of the Composite TD3. I think portions are significant, but it is hard to appreciate from this plot. What might be useful to get a better sense of the data is to include error bars for the results presented in table 2 and table 3. I think table 3 could also benefit with box plots for each of the domains, just to make the comparison easier.

  C7: I think the claim "We also showed that composite TD3 is able to achieve state-of-the-art data-efficiency compared..." is a bit strong, especially given the needed clarifications on the significance of the results and how you set hyperparameters. I would urge the authors to soften this claim, and instead say you provide evidence of composite q-learning's data efficiency as compared to other methods.


  *Other comments not taken into consideration in the review*

  - It was quite difficult to read sections 2 and 3 given how dense they are. I would recommend splitting these sections into multiple paragraphs to make the sections more readable.

-----------
Post discussion/rebuttal:

After reviewing the comments from other reviewers and the discussion with R3, I'm inclined to think this paper could use a bit more work. I think the idea is still interesting and worth pursuing, but given some of the new observations and experiments run the paper needs to make more changes than I would find reasonable for acceptance.

Thanks again for your hard work, and I look forward to seeing this in a future conference.

**Experience Assessment:**

I have published one or two papers in this area.

**Review Assessment: Checking Correctness Of Derivations And Theory:**

I assessed the sensibility of the derivations and theory.

**Review Assessment: Checking Correctness Of Experiments:**

I assessed the sensibility of the experiments.

**Review Assessment: Thoroughness In Paper Reading:**

I read the paper at least twice and used my best judgement in assessing the paper.

---

> ### Author Response · Authors · 2019-11-14
> **Response to Reviewer #2**
>
> We would like to thank the reviewer for the detailed comments. We included most suggestions in the new revision.
>
> C1: We now average over 10 runs and provide two standard deviations and the results of a significance test. The results are highly significant. The submission is updated accordingly.
>
> C2: The lack of clarity was unfortunate. We updated this section in the new version.
>
> S3: We added an additional plot for the Shifted and Truncated Q-values. Since the main difficulty in this task is the temporal horizon, there is no meaningful difference between states.
>
> Q4: We added the default settings of TD3 to the text. While we agree that hyperparameter optimization is indeed very important to get to the full potential of an algorithm, the underlying algorithm in this paper is, in all cases, the same: TD3. The main difference between the approaches lies in the target calculation for Q-learning. Since we wanted to evaluate the influence of the structure of Q-functions on data-efficiency, we did not change crucial hyperparameters such as the learning rate or target updates, since this would lead to another source of potential differences. We assumed hyperparameters for TD3 to be optimized already as we took the settings of the original paper (the same holds for the discount-factor schedule in TD(Delta) or the rollout horizon of MVE-TD3), which we then used for evaluation. However, we evaluated the performance of the baselines for the extended capacity we had to use for the Composite Q-network in the appendix.
>
> S5: Yes, our code is based on the original implementation of TD3 and we acknowleged this in the code submission. We now included a remark also in the text.
>
> Q6: We now provide variance measures in the tables and further included individual comparisons in the appendix.
>
> C7: We updated the submission accordingly.

---

> > ### Comment · AnonReviewer2 · 2019-11-15
> > **Thanks**
> >
> > Thank you for the response.
> >
> > I appreciate the revisions to the main paper, and think these improve the overall quality.
> >
> > After looking through the concerns raised by reviewer one, I'm much less certain of the actual novelty of the contributions. There are also lingering concerns over the claims and experiments run in this paper. Specifically, I had made the assumption you had optimized the parameters for the motivating markov chain and given reviewer one's observations using the code provided this result is much less meaningful than I had originally thought. I had in fact missed that the shifted values were using a separate learning rate, which I agree with reviewer one should be of import in the main paper.
> >
> > I'm still thinking on how the other reviews effects my overall opinion of this paper. I believe the idea is worth thinking about, and could be natural auxiliary tasks (as mentioned by reviewer 1). I also think this may need some more careful study to understand what components are improving performance, and there needs to be a more complete parameter study.

---

> > > ### Author Response · Authors · 2019-11-15
> > > **Re: Thanks**
> > >
> > > "I appreciate the revisions to the main paper, and think these improve the overall quality."
> > >
> > > We thank the reviewer for the positive feedback.
> > >
> > > "After looking through the concerns raised by reviewer one, I'm much less certain of the actual novelty of the contributions."
> > >
> > > As explained in detail in the discussion with Reviewer 3, the original TD-paper indeed includes a general description of predictions for a fixed horizon via TD, however we were the first to precisely formalize said predictions in an off-policy setting. We acknowledge the work of De Asis et al. in the current revision. We would like to point out, however, that we presented initial results in a workshop paper of ours at the RSS 2019 Workshop of Combining Learning and Reasoning – Towards Human-Level Robot Intelligence (https://sites.google.com/view/rss19-learning-and-reasoning) in June 2019, prior to the upload of De Asis et al. The workshop paper is uploaded on their website and also not mentioned in the paper of De Asis et al. The anonymous workshop paper can be found at: https://gofile.io/?c=2Omxmi
> > >
> > > Furthermore, in our definition of Truncated Q-functions, the action-values are w.r.t. to the full return, which is only possible due to the completion by the Shifted Q-function and a major difference to prior work. The main contribution of the paper is the analysis of the interplay between short- and long-term predictions.
> > >
> > > "There are also lingering concerns over the claims and experiments run in this paper. Specifically, I had made the assumption you had optimized the parameters for the motivating markov chain and given reviewer one's observations using the code provided this result is much less meaningful than I had originally thought. I had in fact missed that the shifted values were using a separate learning rate, which I agree with reviewer one should be of import in the main paper."
> > >
> > > We added an analysis of different learning rates for the Shifted Q-functions in Composite Q-learning and Shifted Q-learning in Section 5. "Shifted Q-learning" denotes a definition of the Q-target, where the long-term value is shifted by one time step (i.e. no approximate n-step return). Shifting the value in time alone is slowing down convergence and can be at most as fast as vanilla Q-learning (with a learning rate of 1.0). We hope that this experiment convincingly shows that the speed up is only due to the combination of short- and long-term predictions.
> > >
> > > Furthermore, we added initial results of a shallow architecture of the Q-network (no multi-layered structure) for the Walker2d-v2 environment to the appendix.

---

### Official Review · AnonReviewer3 · 2019-10-23
**Official Blind Review #3**

**Rating:** 1

**Review:**

This paper proposes the Composite Q-learning algorithm, which combines the algorithmic ideas of using compositional TD methods to truncate the horizon of the return, as well as shift a return in time. They claim that this approach will improve the method's data efficiency relative to standard Q-learning. They demonstrate its performance relative to Q-learning in a tabular domain, as well as in deep RL domains which use the compositional idea as an off-policy critic.

Overall, the paper has interesting algorithmic ideas, but there are critical issues in the evaluation and resulting claims being made. Based on this, I am recommending rejection of the paper. I do think there is value in the compositional idea, but for different reasons outlined in the suggestions.

Issues:

1) The truncation of the horizon is not a novel TD formulation, as claimed in the paper. This algorithm is described in the original TD paper (Sutton, 1988) as "prediction by a fixed interval." Sutton's group further has a recent paper following up on the fixed-horizon TD (FHTD) idea (De Asis et al., 2019), introducing an off-policy control variant of it.

2) Based on Theorem 1 of the TD(\Delta) paper (Romoff et al., 2019), as well as the sample-complexity arguments from the FHTD paper, this compositional algorithm is *exactly* equivalent to standard TD in the tabular setting (and function approximation if value functions don't share parameters), update for update, assuming that: (1) each value function is initialized identically, and (2) the same step size is used for each value function. An intuition for why is because the accuracy of the shifted action-values depends on the accuracy of the standard TD estimate, and the TD errors can be shown to exactly decompose that of standard TD. Under this, there is no ready improvement in data efficiency due to the fixed-horizon value functions converging quicker.

3) The results in the tabular setting seem to contradict what I described in Issue 2, because compositional Q-learning as presented did converge quicker than standard Q-learning. However, this is misleading in that the other methods used a step size of 1e-3, but the step size of the shifted value functions used, without explanation, a larger step size of 1e-2. The reason for the improved performance is that these values had a step size an order of magnitude larger than the remaining ones, and if one were to use the same step size across all value functions, it would have matched Q-learning exactly. This exact decomposition is supported by how the fixed-horizon value estimates follow Q-learning's curves exactly for the first h - 1 updates (and will converge to Q-learning's curve if h approaches infinity), and can further be verified by running the provided code with a step size of 1e-3 for the shifted value functions. Without acknowledging the equivalence when using a consistent step size across value functions, as well as sweeping over step sizes for each method, the results don't present a fair comparison and significantly misrepresent compositional TD methods.

4) On this observation that it is an exact decomposition of TD, it is particularly an exact decomposition of *one-step* TD, as one-step TD errors are used in the fixed-horizon and shifted value function estimates. This makes it equally biased to a one-step method, and is inconsistent with the use of "multi-step" learning in the literature where information across several time steps is included in the estimate of the return. Truncating and shifting things in time can be contextualized as a form of time-dependent discounting, and adjusting the discount rate isn't generally viewed as performing multi-step TD.

5) Based on the above, the benefit in the deep RL setting is not convincingly due to what is claimed (as parameters are shared, and a consistent step size is used in the optimizer). Some possible reasons might include the architectural choices in how the network represented the decomposition, as well as the representation learning benefits of predicting many relevant outputs to a task.

Suggestions:

1) The precise novelty of the work can be clarified, as the fixed-horizon TD formulation dates back to Sutton (1988), and has been extensively studied in De Asis et al. (2019). As far as I'm aware, there's novelty in the idea of shifting value functions, reconstructing the full return from decomposed value functions, and introducing a penalty to the loss based on inconsistencies in the value estimates.

2) The motivation and claims of the paper should be revised, as the claimed data efficiency from fixed-horizon values converging quicker isn't readily true. The resulting deep RL results may need more careful experiments to tease apart why the composition might be helping. For example, it might be useful to compare a different neural network architecture, like having all of the compositional components as outputs from the same, final hidden layer (in comparison with outputting them from intermediate hidden layers).

3) The tabular example needs to be re-worked to ensure a fair comparison between each algorithm. For example, the curves can be presented under the best step size (in terms of some metric, like area under the curve) for each algorithm. While there is an exact equivalence to standard one-step TD methods, a real benefit of the approach is that strictly more information is present to the agent, and the flexibility of being able to use separate step sizes for each value function can be favorable if it can be shown to be better after fairly tuning each algorithm. Shifting the focus toward showing that certain types of value functions are less sensitive to step sizes or work better operating at different time-scales from other components (because this seems to be what's actually happening in the results) would be a huge plus for this.

4) Because it is using one-step TD errors to estimate each of these components, and is equally biased to one-step TD, it isn't really a multi-step method. I think it would be better to emphasize the compositional aspect and its increased flexibility, than frame it as a multi-step off-policy method.

----------

Post-rebuttal:

I think the additional results post-discussion are good, and are on the right track of the claimed goal of analyzing the algorithm. However, the new results might be contradictory to some of the claims made earlier in the paper, and so a more involved revision seems to be needed. I do believe the algorithm has promise for the reasons teased apart in our discussion, and encourage the authors to improve their paper with these results.

To detail a few things:

1) The new results, which now empirically demonstrate the exact equivalence with one-step Q-learning, contradicts some claims about improved data efficiency due to truncated value-functions converging quicker. While meta-parameter selection isn't the focus, if the choice of meta-parameter is what can make it differ from vanilla Q-learning, and is the key explanation for the improvements, then the analysis should focus on this.

2) Mention of the equivalence only comes up in the experimental results, when it's a key property of the algorithm. If analysis of the composition is the paper's focus, acknowledging this property is foundational to any analysis of the method. It could have been shown analytically following the algorithm's derivation, and would have better justified some of the choices made in the experiments.

3) Being equivalent to running *one-step* Q-learning still makes the "multi-step" learning emphasis appear incorrect, especially when the algorithm can trivially be extended to use actual multi-step TD methods. The title seems to come from interpreting what the composite values represent, but the horizon isn't what makes a method multi-step, and the compositional components add up to exactly one-step Q-learning's update.

Minor:

1) Arguably one of the most prevalent explanations in the deep RL literature for why one might expect improvements is the multi-task/auxiliary task hypothesis (Jaderberg et al., 2016).

**Experience Assessment:**

I have published one or two papers in this area.

**Review Assessment: Checking Correctness Of Derivations And Theory:**

I carefully checked the derivations and theory.

**Review Assessment: Checking Correctness Of Experiments:**

I carefully checked the experiments.

**Review Assessment: Thoroughness In Paper Reading:**

I read the paper thoroughly.

---

> ### Author Response · Authors · 2019-11-14
> **Response to Reviewer #3**
>
> First of all, we would like to thank the reviewer for the extensive and valuable feedback.
>
> Suggestion 1) While we acknowledge the work of De Asis et al., we would like to mention that we presented initial results in a workshop paper of ours at the RSS 2019 Workshop of Combining Learning and Reasoning – Towards Human-Level Robot Intelligence (https://sites.google.com/view/rss19-learning-and-reasoning) in June 2019, prior to the upload of De Asis et al. The workshop paper is uploaded on their website and also not mentioned in the paper of De Asis et al. The anonymous workshop paper can be found at: https://gofile.io/?c=2Omxmi
>
> Furthermore, the formulation in FHTD has a small yet critical difference to the truncated formulation in our submission. The maximizing action in FHTD is according to the truncated value-function of the former step, not w.r.t. the full return as in our work. Taking the full return is only possible due to the completion based on the Shifted Q-function and is a major difference to prior work. As suggested by the reviewer, however, we added a more precise formulation of the contributions in the abstract, in the introduction and in related work.
>
> Suggestion 2) We added a first comparison between our architecture and a shallow Composite Q-network for the Walker2d-v2 environment in the appendix. We will add results for the other environments in the camera-ready version.
>
> Issue 3) Shifting the value function to overcome the necessity of a model indeed imposes a bottleneck which has to be tackled by either generalization (as in the TD3 case) or by adjusting the learning rate. We would like to point out that the full value function is still updated with the same step size to its given target in all approaches. The Shifted Q-function will always be updated slower than the true value function which is why we still believe this to be a fair comparison. We added an explanation in Sections 4 and 5.
>
> Issue 4) We would like to refer to the lower row of Fig. 4, where we compare the different TD-errors over time. Please note, that the TD-errors for the truncated Q-functions are lower across the whole of training and decrease with shorter horizons (being the least for Tr_0). Therefore, the targets for some horizon h, bootstrapping from horizon h-1, are less biased -- grounded by the target of Tr_0, which has zero bias. While we can expect the Shifted Q-approximation to be similarly biased as the full Q-approximation, the first part of the target for the full Q-estimation, which has an even higher weight due to discounting, is less biased. We can therefore consider Composite Q-learning a bias reduction technique. The price is an increase in variance which is the reason for our novel regularization technique.
>
> Suggestion 3) We thank the reviewer for the suggestion and consider the hyperparameter optimization of learning rates as an extension for the camera-ready version. However, our main focus was not on maximum performance, but on the analysis of the structure of Q-functions.
>
> Suggestion 4) We agree that the single term "Multi-step" alone usually refers to an unbiased sum of real consecutive rewards in the literature. Since our approach includes off-policy approximations of n-step returns within target calculation for Q-learning (greedy target policy), we argue that Composite Q-learning belongs to this area of research -- also with respect to the reasons outlined in detail above.

---

> > ### Comment · AnonReviewer3 · 2019-11-14
> > **Re: Response to Reviewer #3**
> >
> > "Furthermore, the formulation in FHTD has a small yet critical difference to the truncated formulation in our submission. The maximizing action in FHTD is according to the truncated value-function of the former step, not w.r.t. the full return as in our work. Taking the full return is only possible due to the completion based on the Shifted Q-function and is a major difference to prior work. As suggested by the reviewer, however, we added a more precise formulation of the contributions in the abstract, in the introduction and in related work."
> >
> > Thank you for the clarification, and the timeline. Despite that, Sutton's original TD paper still describes the overall procedure of the consecutive bootstrapping for estimating these quantities. Of note, De Asis et al. (2019) still provide analysis motivating the use of truncating the horizon with function approximation. The completion of the return is indeed a major difference, but what makes it possibly lose the theoretical benefits of the truncated values.
> >
> > "Issue 3) Shifting the value function to overcome the necessity of a model indeed imposes a bottleneck which has to be tackled by either generalization (as in the TD3 case) or by adjusting the learning rate. We would like to point out that the full value function is still updated with the same step size to its given target in all approaches. The Shifted Q-function will always be updated slower than the true value function which is why we still believe this to be a fair comparison. We added an explanation in Sections 4 and 5."
> >
> > As to why this is not readily a fair comparison, because composite Q-learning exactly decomposes one-step Q-learning, it's left to justify that the extra information available to the agent can be used to learn quicker. One can use the code that's provided and find a larger step size for vanilla Q-learning which makes it outperform *every method* presented in the figure, that it's not definitively shown that the composition is helping- I do believe that it can be shown, but the results as presented do not show this.
> >
> > "Issue 4) We would like to refer to the lower row of Fig. 4, where we compare the different TD-errors over time. Please note, that the TD-errors for the truncated Q-functions are lower across the whole of training and decrease with shorter horizons (being the least for Tr_0). Therefore, the targets for some horizon h, bootstrapping from horizon h-1, are less biased -- grounded by the target of Tr_0, which has zero bias. While we can expect the Shifted Q-approximation to be similarly biased as the full Q-approximation, the first part of the target for the full Q-estimation, which has an even higher weight due to discounting, is less biased. We can therefore consider Composite Q-learning a bias reduction technique. The price is an increase in variance which is the reason for our novel regularization technique."
> >
> > The lower TD errors for the truncated horizons are not indicative of the bias of the action-values corresponding to the complete return, which is what's being used for decision making. They have a different target they are estimating, and while those action-values themselves have lower-biased estimators, they *exactly* decompose the bias of the estimate for the complete return. As such, the bias of the complete return has not been decreased. If one were to run the provided code under the same random seeds, with the same step size across all value functions, it can be seen that the curves are *exactly* equivalent (apart from perhaps floating point errors).
> >
> > "Suggestion 3) We thank the reviewer for the suggestion and consider the hyperparameter optimization of learning rates as an extension for the camera-ready version. However, our main focus was not on maximum performance, but on the analysis of the structure of Q-functions."
> >
> > When one is claiming that an algorithm is better than another, then hyperparameter optimization can't really be avoided. However, it's less about optimization and more so that such analysis of the structure requires analyzing how hyperparameters interplay. Part of this suggestion is that the analysis of the structure seems to be missing, as it failed to acknowledge (1) the exact decomposition of one-step TD, and (2) justifying the use of a larger step size for the shifted action-values.
> >
> > "Suggestion 4) We agree that the single term "Multi-step" alone usually refers to an unbiased sum of real consecutive rewards in the literature. Since our approach includes off-policy approximations of n-step returns within target calculation for Q-learning (greedy target policy), we argue that Composite Q-learning belongs to this area of research -- also with respect to the reasons outlined in detail above."
> >
> > It does approximate an n-step return, in a sense that *one-step* Q-learning approximates an *infinite-step* return. This is misleading as one can still use actual multi-step TD methods to estimate the truncated and shifted values.

---

> > > ### Author Response · Authors · 2019-11-14
> > > **Re: Re: Response to Reviewer #3**
> > >
> > > "Thank you for the clarification, and the timeline. Despite that, Sutton's original TD paper still describes the overall procedure of the consecutive bootstrapping for estimating these quantities. Of note, De Asis et al. (2019) still provide analysis motivating the use of truncating the horizon with function approximation. The completion of the return is indeed a major difference, but what makes it possibly lose the theoretical benefits of the truncated values."
> > >
> > > It does in a very general manner, which is now acknowledged in related work. The benefits of bootstrapping from different greedy policies for different horizons, as in De Asis et al., come at the cost of not necessarily being optimal w.r.t. the complete task. De Asis et al. dismiss this drawback by stating that approximations always suffer from impreciseness: "For a final horizon H << infinity, there may be concerns about suboptimal control. We explore this empirically in Section 5. For now, we note that optimality is never guaranteed when values are approximated." [1]
> > >
> > > "As to why this is not readily a fair comparison, because composite Q-learning exactly decomposes one-step Q-learning, it's left to justify that the extra information available to the agent can be used to learn quicker. One can use the code that's provided and find a larger step size for vanilla Q-learning which makes it outperform *every method* presented in the figure, that it's not definitively shown that the composition is helping- I do believe that it can be shown, but the results as presented do not show this."
> > >
> > > The main difficulty of the given MDP is simply the horizon and it is designed to be that way on purpose -- to rule out any other source of difference except for the horizon. Even though we do understand the point being made, the full Q-values are still updated with the same learning rate for all approaches, regardless of their given targets. Shifting alone would not lead to a speed up. This comes only due to the combination with the Truncated Q-function. The step size could be set to almost 1 for the given MDP and all approaches, which is due to the simplicity of the problem. There still is a significant difference in convergence for the given fixed learning rate which can only be explained by the combination of Truncated and Shifted Q-values. The approach, however, does have the limitation, that one can only expect a benefit, if there is generalization among states (as in the TD3-experiments with the given multi-layered architecture) or if the learning rate for the Shifted Q-function can be set to a higher value. This is indeed acknowledged in the current revision.
> > >
> > > "The lower TD errors for the truncated horizons are not indicative of the bias of the action-values corresponding to the complete return, which is what's being used for decision making. [...] It does approximate an n-step return, in a sense that *one-step* Q-learning approximates an *infinite-step* return."
> > >
> > > With the difference of being grounded by the unbiased immediate reward as a target for Tr_0. The targets in Composite Q-learning represent a sum of partial sums of length n, each with a lower bias (according to the lower row in Fig. 4) -- in contrast to vanilla Q-learning which bootstraps from the long-term prediction for every time step.
> > >
> > > [1] Fixed-Horizon Temporal Difference Methods for Stable Reinforcement Learning, De Asis et al., 2019. https://arxiv.org/abs/1909.03906

---

> > > > ### Comment · AnonReviewer3 · 2019-11-14
> > > > **Re: Re: Re: Response to Reviewer #3**
> > > >
> > > > "Shifting alone would not lead to a speed up... There still is a significant difference in convergence for the given fixed learning rate which can only be explained by the combination of Truncated and Shifted Q-values"
> > > >
> > > > From setting the step size of the shifted value functions to 1e-3, it can be shown that the faster shifting *is* why it is being sped up. The provided code can be run with a step size of 1e-2 for the shifted value functions (what's presented in the paper), and 1e-3 for the shifted value functions, and see that that change is what's making it learn faster. My guess as to why would be that because shifted values treat the immediate reward as 0, they only have to average out the variability in the next state when estimating the expectation, and thus can tolerate operating at a quicker timescale.
> > > >
> > > > The composition is what lets the full return take advantage of this faster shifting, but the composition alone when using 1e-3 for *every* value function does not speed it up. Allowing for this flexibility is a real benefit of the method even in a simple setting, but it needs to be shown with a more focused analysis (either theoretical or empirical).
> > > >
> > > > "With the difference of being grounded by the unbiased immediate reward as a target for Tr_0. The targets in Composite Q-learning represent a sum of partial sums of length n, each with a lower bias (according to the lower row in Fig. 4) -- in contrast to vanilla Q-learning which bootstraps from the long-term prediction for every time step."
> > > >
> > > > Beyond the 1-step horizon, these are biased estimates of the n-step sums of rewards (which accumulate bias along the bootstrapping chain), and will further add in the bias of the shifted values. Without parameter sharing, and with the same step size for all value functions, these not only add up to be equally biased to running one-step Q-learning, but add up to the exact same update as one-step Q-learning (i.e., as if one were bootstrapping from the long-term prediction on every time step).
> > > >
> > > > "The approach, however, does have the limitation, that one can only expect a benefit, if there is generalization among states (as in the TD3-experiments with the given multi-layered architecture) or if the learning rate for the Shifted Q-function can be set to a higher value. This is indeed acknowledged in the current revision."
> > > >
> > > > I appreciate this revision, but a concern is that because these are the conditions where one could expect a benefit, this paper should focus its analysis on this- the part about expecting a benefit from the shifted values being set to a higher value isn't discussed anywhere. As a side note that may be of interest, beyond expecting benefits from generalization, another possibility in the deep RL setting that's reasonably acknowledged in the literature is the representation learning benefits from predicting many relevant outputs to a task [1].
> > > >
> > > > [1] https://arxiv.org/abs/1611.05397

---

> > > > > ### Author Response · Authors · 2019-11-15
> > > > > **Re: Re: Re: Re: Response to Reviewer #3**
> > > > >
> > > > > "From setting the step size of the shifted value functions to 1e-3, it can be shown that the faster shifting *is* why it is being sped up. The provided code can be run with a step size of 1e-2 for the shifted value functions (what's presented in the paper), and 1e-3 for the shifted value functions, and see that that change is what's making it learn faster. My guess as to why would be that because shifted values treat the immediate reward as 0, they only have to average out the variability in the next state when estimating the expectation, and thus can tolerate operating at a quicker timescale.
> > > > >
> > > > > The composition is what lets the full return take advantage of this faster shifting, but the composition alone when using 1e-3 for *every* value function does not speed it up. Allowing for this flexibility is a real benefit of the method even in a simple setting, but it needs to be shown with a more focused analysis (either theoretical or empirical)."
> > > > >
> > > > > We would like to thank the reviewer for the effort in reviewing this paper and the fruitful discussion and suggestions.
> > > > >
> > > > > We added an analysis of the learning rate for the Shifted Q-functions in Composite Q-learning and Shifted Q-learning to the discussion of the tabular setting. We denote by "Shifted Q-learning" a definition of the Q-target, where the long-term value is shifted by one time step (i.e. no approximate n-step return). One can see that shifting alone does not lead to faster convergence, even when setting the learning rate of the Shifted Q-function to 1. In fact, shifting the value in time is slowing down convergence. We hope that this experiment convinces the reviewer that the faster convergence can only be explained by the interplay of Shifted and Truncated Q-functions.
> > > > >
> > > > > "I appreciate this revision, but a concern is that because these are the conditions where one could expect a benefit, this paper should focus its analysis on this- the part about expecting a benefit from the shifted values being set to a higher value isn't discussed anywhere."
> > > > >
> > > > > Besides the new evaluation in Section 5.1, we further added an initial comparison of our architecture to a shallow network in the appendix.

---

> > > > > > ### Comment · AnonReviewer3 · 2019-11-15
> > > > > > **Re: Re: Re: Re: Re: Response to Reviewer #3**
> > > > > >
> > > > > > "We added an analysis of the learning rate for the Shifted Q-functions in Composite Q-learning and Shifted Q-learning to the discussion of the tabular setting. We denote by "Shifted Q-learning" a definition of the Q-target, where the long-term value is shifted by one time step (i.e. no approximate n-step return). One can see that shifting alone does not lead to faster convergence, even when setting the learning rate of the Shifted Q-function to 1. In fact, shifting the value in time is slowing down convergence."
> > > > > >
> > > > > > Thank you for generating the additional figure. I apologize for any miscommunication, but when suggesting that the faster shifting is why it is being sped up, I wasn't referring to "Shifted Q-learning" where one is learning the shifted target alone without any interplay in the composition. I was referring to faster shifting *within* composite Q-learning, the green curves in the figure, and not the purple ones. As a side note, I think the figure should perhaps show "Composite Q-learning (10−3)" or "Q-learning" with dashed lines, as "Composite Q-learning (10−3)" is not visible from being directly beneath Q-learning.
> > > > > >
> > > > > > "We hope that this experiment convinces the reviewer that the faster convergence can only be explained by the interplay of Shifted and Truncated Q-functions."
> > > > > >
> > > > > > Like the new figure suggests with the green lines in comparison with the yellow line, as only the step size of the shifted action-values is being varied in composite Q-learning, the faster shifting *within* composite Q-learning is what's speeding it up. Something which can further attribute the benefits to the faster shifting is that if one were to use a larger step size for the truncated values (while keeping the step sizes of the composed values and shifted values at 1e-3), it doesn't have nearly as large an improvement, and sometimes does *worse* from plateauing at a poor steady-state error!
> > > > > >
> > > > > > "Composite Q-learning (10^−3)" is also using the composition, but from the exact equivalence when using the same step size for all value functions within the composition, the composition and its interplay does not readily improve any data efficiency or decrease the bias in the estimate. This is what is meant by the composition alone not being responsible for speeding things up, and not readily taking advantage of truncated action-values converging quicker. The bias isn't readily being decreased because the truncated values have less-biased targets, but because an internal component is using a larger step size, which decreases bias quicker than a smaller step size, depending on the variability in the target. The result using 1e-3 for all components contradicts that the benefit is "only due to the combination of short- and long-term predictions," but is in support that it is due to a combination of slow and fast timescale predictions within the composition. Put another way, an intuition might be that faster shifting is what lets one compose it with the truncated action-values earlier than usual, and from then on accelerates via closed-loop feedback.
> > > > > >
> > > > > > Enabling and using this larger "internal" step size for the shifting component (within the interplay) appears to be the key benefit of the composition, a positive result supported by your own results and figures. This result currently exists in the text as an acknowledgement that a step size of 1e-2 was used for the shifted values. The text lacks an explanation for why one might want to use a larger step size for this component, and why it can tolerate using a larger step size in general, as a larger step size for the truncated values don't provide nearly as much benefit. With the new figure, some results are there for supporting this choice, but the text should discuss and emphasize this!

---

> > > > > > > ### Author Response · Authors · 2019-11-15
> > > > > > > **Re: Re: Re: Re: Re: Re: Response to Reviewer #3**
> > > > > > >
> > > > > > > We are grateful for the very much non-standard and constructive efforts of the reviewer.
> > > > > > >
> > > > > > > "Thank you for generating the additional figure. I apologize for any miscommunication, but when suggesting that the faster shifting is why it is being sped up, I wasn't referring to "Shifted Q-learning" where one is learning the shifted target alone without any interplay in the composition. I was referring to faster shifting *within* composite Q-learning, the green curves in the figure, and not the purple ones. As a side note, I think the figure should perhaps show "Composite Q-learning (10−3)" or "Q-learning" with dashed lines, as "Composite Q-learning (10−3)" is not visible from being directly beneath Q-learning."
> > > > > > >
> > > > > > > We will change this for the camera-ready.
> > > > > > >
> > > > > > > "Like the new figure suggests with the green lines in comparison with the yellow line, as only the step size of the shifted action-values is being varied in composite Q-learning, the faster shifting *within* composite Q-learning is what's speeding it up. Something which can further attribute the benefits to the faster shifting is that if one were to use a larger step size for the truncated values (while keeping the step sizes of the composed values and shifted values at 1e-3), it doesn't have nearly as large an improvement, and sometimes does *worse* from plateauing at a poor steady-state error!"
> > > > > > >
> > > > > > > We thank the reviewer for the input. We are currently evaluating the counterpart, changing the learning rates for the Truncated Q-functions while keeping the learning rates for the full Q-estimate and the Shifted Q-functions fixed. Since the deadline of the rebuttal is coming close, we will provide the results in the camera-ready latest.
> > > > > > >
> > > > > > > "Enabling and using this larger "internal" step size for the shifting component (within the interplay) appears to be the key benefit of the composition, a positive result supported by your own results and figures. This result currently exists in the text as an acknowledgement that a step size of 1e-2 was used for the shifted values. The text lacks an explanation for why one might want to use a larger step size for this component, and why it can tolerate using a larger step size in general, as a larger step size for the truncated values don't provide nearly as much benefit. With the new figure, some results are there for supporting this choice, but the text should discuss and emphasize this!"
> > > > > > >
> > > > > > > We agree and these points were exactly what we tried to illustrate with our experiments as well. The benefit of the interplay of the components depends on having the two decoupled parts learned on two time scales via different learning rates or generalization (as in the deep RL experiments). We are happy to see the large common ground of understanding of the problems and will try to add further clarifications in the camera-ready version.

---

> > > > > > > > ### Author Response · Authors · 2019-11-15
> > > > > > > > **Re: Re: Re: Re: Re: Re: Response to Reviewer #3**
> > > > > > > >
> > > > > > > > We updated Section 5.1 and included a new evaluation of the learning rates of the Truncated Q-functions. We again would like the reviewer for the fruitful discussions.

---

> > > > > > > > > ### Comment · AnonReviewer3 · 2019-11-15
> > > > > > > > > **Re: Re: Re: Re: Re: Re: Re: Response to Reviewer #3**
> > > > > > > > >
> > > > > > > > > Thank you for the revision- these indeed paint a more complete picture of the algorithm. I think it might still be worthwhile to justify the choice of fixing the step size of 1e-3 for the values representing the full return, as with many TD methods, step size sensitivity can considerably vary. Acknowledging the exact decomposition of one-step Q-learning though, a quick argument for this choice would be that composite Q-learning could never do worse than one-step Q-learning if it can be set to match it exactly. :)
> > > > > > > > >
> > > > > > > > > While most of this has been about the tabular domain, this is where things are conceptually clear and can be quickly, and reasonably confidently, teased apart to (1) justify the choice of the larger step size used in Figure 2, and (2) make a case for what might be happening in the deep RL setting (or what to try next!). For example, drawing intuition from the tabular results, it may be possible that the architectural choice of outputting the value function components components at earlier hidden layers are interpretable as implicitly running things at different time scales- changes to earlier hidden layers might take longer for something further down the network to adapt. However, this is harder to verify due to other dependencies like hidden layer sizes, activation functions, optimizer, etc.

---

### Author Response · Authors · 2019-11-14
**Overview of changes in the new revision**

We would like to thank all reviewers for the constructive feedback and for their efforts in reviewing this paper.

We uploaded a new revision including most suggestions of the reviewers.

The main changes are:
1) A more precise formulation of the contributions
2) A more detailed explanation for the results in the tabular setting and average performance over multiple runs
3) Variance measures for the results in Tables 2 and 3
4) More experiments with different settings of the regularization weight and a shallow Q-network architecture
5) A new evaluation of different learning rates for the Shifted Q-functions in the tabular setting
6) A new evaluation of different learning rates for the Truncated Q-functions in the tabular setting

---

### Decision · Program_Chairs · 2019-12-19

**Decision:**

Reject

**Comment:**

The authors propose TD updates for Truncated Q-functions and Shifted Q-functions, reflecting short- and long-term predictions, respectively. They show that they can be combined to form an estimate of the full-return, leading to a Composite Q-learning algorithm. They claim to demonstrated improved data-efficiency in the tabular setting and on three simulated robot tasks.

All of the reviewers found the ideas in the paper interesting, however, based on the issues raised by Reviewer 3, everyone agreed that substantial revisions to the paper are necessary to properly incorporate the new results. As a result, I am recommending rejection for this submission at this time. I encourage the authors to incorporate the feedback from the reviewers, and believe that after that is done, the paper will be a strong submission.